# Emerging role of oncogenic ß-catenin in exosome biogenesis as a driver of immune escape in hepatocellular carcinoma

Camille Dantzer[1], Justine Vaché[1], Aude Brunel[2], Isabelle Mahouche[1], Anne-Aurélie Raymond[1,3], Jean-William Dupuy[3,4], Melina Petrel[5], Paulette Bioulac-Sage[1], David Perrais[6], Nathalie Dugot-Senant[7], Mireille Verdier[2], Barbara Bessette[2], Clotilde Billottet[1*†], Violaine Moreau[1*†]

[1]Université de Bordeaux, INSERM, U1312, BRIC, Bordeaux, France; [2]Université de Limoges, INSERM, U1308, CAPTuR, Limoges, France; [3]Plateforme OncoProt, Université de Bordeaux, CNRS, INSERM, TBM-Core, US5, UAR3457, Bordeaux, France; [4]Plateforme Protéome, Université de Bordeaux, Bordeaux Proteome, Bordeaux, France; [5]Bordeaux Imaging Center, Université de Bordeaux, CNRS, INSERM, BIC, Bordeaux, France; [6]Université de Bordeaux, CNRS, Interdisciplinary Institute for Neuroscience, IINS, Bordeaux, Bordeaux, France; [7]Plateforme d'histologie, Université de Bordeaux, CNRS, INSERM, TBM-Core, US5, UAR3457, Bordeaux, France

*For correspondence:
clotilde.billottet@u-bordeaux.fr (CB);
violaine.moreau@inserm.fr (VM)

†These authors contributed equally to this work

Competing interest: The authors declare that no competing interests exist.

**Abstract** Immune checkpoint inhibitors have produced encouraging results in cancer patients. However, the majority of ß-catenin-mutated tumors have been described as lacking immune infiltrates and resistant to immunotherapy. The mechanisms by which oncogenic ß-catenin affects immune surveillance remain unclear. Herein, we highlighted the involvement of ß-catenin in the regulation of the exosomal pathway and, by extension, in immune/cancer cell communication in hepatocellular carcinoma (HCC). We showed that mutated ß-catenin represses expression of *SDC4* and *RAB27A*, two main actors in exosome biogenesis, in both liver cancer cell lines and HCC patient samples. Using nanoparticle tracking analysis and live-cell imaging, we further demonstrated that activated ß-catenin represses exosome release. Then, we demonstrated in 3D spheroid models that activation of β-catenin promotes a decrease in immune cell infiltration through a defect in exosome secretion. Taken together, our results provide the first evidence that oncogenic ß-catenin plays a key role in exosome biogenesis. Our study gives new insight into the impact of ß-catenin mutations on tumor microenvironment remodeling, which could lead to the development of new strategies to enhance immunotherapeutic response.

## eLife assessment

Hepatocellular carcinoma (HCC) is a particularly aggressive form of cancer, with an increasing number of treatment options approved for use in patients over the past decade. However, the biology of HCC and identifiable therapeutic targets have not been as clear, even in the era of molecular oncology. Likewise, the cellular biology of HCC, including the role of intercellular communication, has not been well elucidated. In this **compelling** study, Dantzer et al. provide **fundamental** insight into the role of beta-catenin on intercellular communication occurring via extracellular vesicles, with implications for immune evasion in a cancer increasingly being treated using immuno-oncologic agents.

## Introduction

HCC is the most frequent form of primary liver adult cancer (80% of cases). HCC represents the sixth most commonly diagnosed cancer worldwide and the third leading cause of cancer-related death (*Sung et al., 2021*). This pathology has a poor prognosis and the diagnosis is often made at an advanced stage for which treatment options are limited, especially if surgery is no longer possible. In 2020, the combination of Atezolizumab (anti-PD-L1) and Bevacizumab (anti-VEGF) became the first-line FDA-approved therapy for advanced HCC (*Finn et al., 2020*). Despite this therapeutic advance, several studies emphasized that ß-catenin-mutated HCC are devoid of immune infiltrates and are resistant to immunotherapy (*Akasu et al., 2021*; *Pinyol et al., 2019*; *Ruiz de Galarreta et al., 2019*; *Sia et al., 2017*). *CTNNB1* gene mutations are found in 30–40% of HCC and trigger uncontrolled transcriptional activity (*de La Coste et al., 1998*; *Rebouissou et al., 2016*). These mutations prevent β-catenin degradation, fostering its role as a transcriptional co-factor and thus the expression of genes involved in cell proliferation and invasion. This altered pattern of gene expression could provide a key to understanding to the observed immune evasion. ß-catenin, encoded by the *CTNNB1* gene, is a main oncogene, mutated in various cancers, such as melanoma and liver, endometrial, and colorectal cancers (*Kim and Jeong, 2019*). These ß-catenin-mutated tumors share specific features, prominent among them a microenvironment devoid of immune infiltrates. Thus, despite of a therapeutic revolution for cancer treatment, with the emergence of immunotherapy and more particularly immune checkpoint inhibitors (anti-PD1, anti-PD-L1), most of the ß-catenin-mutated tumors remain resistant to immunotherapy (*Yaguchi et al., 2012*; *Spranger et al., 2015*; *Du et al., 2020*; *Cen et al., 2021*; *Rotman et al., 2020*; *Wang et al., 2020*). In these tumors, the oncogenic β-catenin is able to establish a microenvironment that favors tumor progression notably by promoting immune escape. Few studies have reported that the oncogenic β-catenin is implicated in the impairment of intercellular communication between cancer cells and immune cells, partly through soluble molecules such as cytokines. In melanoma and HCC, a decrease in CCL4 and CCL5, leads to a defective recruitment of dendritic cells and consequently impaired T-cell activity and immune escape (*Ruiz de Galarreta et al., 2019*; *Spranger et al., 2015*). As immunotherapies are now part of the growing arsenal for treating cancers, it is important to better understand the mechanisms underlying this immune escape phenotype. In this study, we focused on ß-catenin-mutated HCC and intercellular communication through extracellular vesicles.

Tumor-derived extracellular vesicles (EVs) are increasingly described as important actors in tumor microenvironment communication (*van Niel et al., 2018*); but no studies explored their role in the immune escape and immunotherapy resistance of ß-catenin-mutated tumors. EVs are nanometric phospholipid bilayer transport vesicles, which contain various cargoes such as RNAs or proteins and that impact the properties and functions of recipient cells. EVs can be classified into two main categories: microvesicles and exosomes. Microvesicles (50–1000 nm) correspond to EVs directly secreted from the plasma membrane (PM). By contrast, exosomes (50–150 nm) are released from an endosomal compartment, the multivesicular body (MVB), by its fusion with the cell surface (*van Niel et al., 2018*). Among the actors implicated in exosome biogenesis and secretion, syndecan-4, involved in the endosomal membrane budding (*Baietti et al., 2012*) and Rab27a, which promotes the docking of MVB to the PM, play key roles in exosome release (*Ostrowski et al., 2010*). As important mediators of cell-to-cell communication, EVs can regulate tumor growth, angiogenesis, invasion, and infiltration of immune cells into the tumor (*Tkach and Théry, 2016*). In the context of HCC, it was reported that EVs derived from tumor cells can modulate epithelial-to-mesenchymal transition, intrahepatic metastasis, and tumor immunity (*Lee et al., 2021*).

Based on these considerations, the current study investigated the role of exosomes in mutated ß-catenin-mediated immune evasion in HCC. We demonstrated that mutated ß-catenin represses syndecan-4 and Rab27a expression, thus altering exosomal secretion from HCC cells, in turn leading to a defective recruitment of immune cells in tumors.

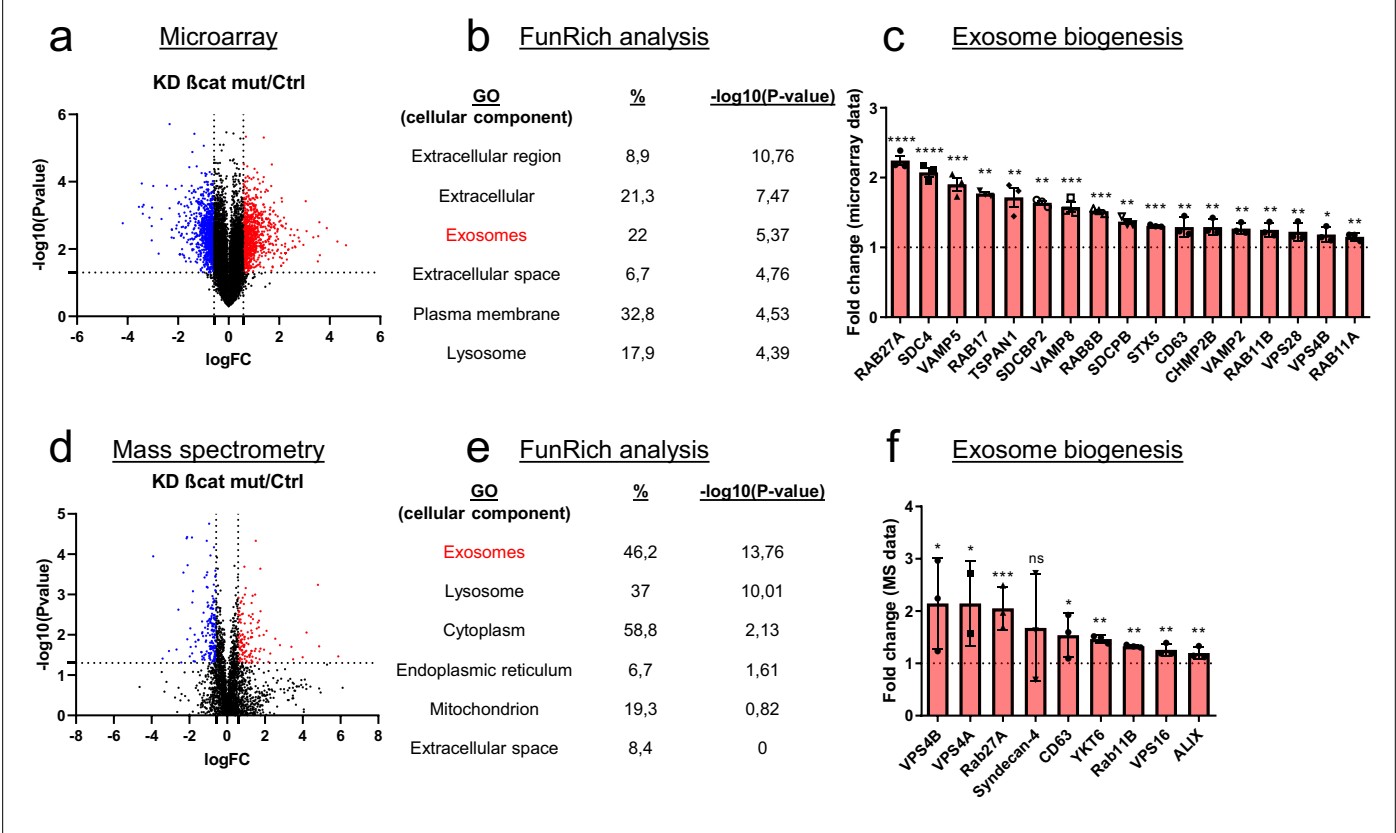

**Figure 1.** Mutated ß-catenin regulates exosome biogenesis gene expression in HepG2 cells. (**a–f**) HepG2 cells were transfected with siRNAs targeting mutated ß-catenin or control siRNAs. (**a**) Volcano plot of deregulated genes identified by microarray based-transcriptomic analysis. Red and blue dots indicated respectively significantly up- and down-regulated genes. (**b**) Upregulation of cellular component genes using FunRich software. (**c**) Upregulated genes associated with exosome biogenesis pathway. The graph indicates the fold change (FC) when comparing the mutated ß-catenin silencing condition with the control condition. (**d**) Volcano plot of deregulated proteins identified by mass spectrometry. Red and blue dots indicated respectively significantly up- and down-regulated proteins. (**e**) Upregulation of cellular component proteins identified using FunRich software. (**f**) Upregulated proteins associated with exosome biogenesis pathway. The graph indicates the fold change when comparing the mutated ß-catenin silencing condition with the control condition. Results are expressed as Mean ± SEM, two-tailed Student's t-test analysis. *p<0.05; **p<0.01; ***p<0.001; ****p<0.0001; ns, non-significant.

## Results

### Silencing of mutated ß-catenin increases exosome biogenesis-associated gene expression in HepG2 cells

*CTNNB1* mutations in HCC are often monoallelic, leaving a wild-type allele in tumor cells. To study the oncogenic ß-catenin specifically, we made use of the dual ß-catenin knockdown (KD) HepG2 model we published recently (*Gest et al., 2023*). The transcriptional analysis of HepG2 KD cells revealed that the expression of 1973 genes (log2(Fold-change) >0.5 and<−0.5) is modulated upon mutated ß-catenin silencing (*Gest et al., 2023*; *Figure 1a*). Gene ontology analysis (FunRich) performed on upregulated genes showed that mutated ß-catenin silencing enhanced significantly the expression of genes linked to specific cellular components, including exosomes (*Figure 1b*). Moreover, silencing of the mutated ß-catenin led to the overexpression of 17 genes associated with exosome biogenesis, such as *RAB27A* and *SDC4* (*Figure 1c*). Similarly, proteomic analysis of HepG2 KD cells revealed exosomes as the main cellular component displaying up-regulated proteins upon mutated ß-catenin silencing (*Figure 1d–e*). Consistent with the transcriptomic analysis, we found Rab27a and Syndecan-4 among up-regulated proteins (*Figure 1f*), even if the variability was found for Syndecan-4 protein levels. These results suggest the involvement of mutated ß-catenin in the regulation of the exosomal pathway.

## Stable depletion of mutated ß-catenin increases exosome secretion in the HepG2 cell model

In the model presented above, siRNAs were used to transiently silence mutated ß-catenin expression in HepG2 cells (*Gest et al., 2023*). To improve this method, the same sequences were used to develop an inducible shRNA strategy to stably inhibit the expression of the mutated ß-catenin. Through the use of doxycycline-inducible promoters, we were able to regulate the expression of shRNAs over time for more in-depth and specific investigations of ß-catenin function in various cellular processes. In HepG2 cells, this strategy reduced protein expression by 62% for mutated ß-catenin and by 42% for CyclinD1, a ß-catenin-positive transcriptional target (*Figure 2a*). The model was also validated regarding mRNA expression of several ß-catenin transcriptional targets. We noted a decrease in the expression of *CCND1* and *AXIN2* (positive targets) and an increase in the expression of *ARG1* expression (negative target) (*Figure 2—figure supplement 1*). Also, as previously described with the siRNA strategy (*Gest et al., 2023*), after mutated ß-catenin depletion in HepG2 cells, we observed an increase in the number and size of bile canaliculi (BC), feature of more differentiated cells (*Figure 2—figure supplement 2*). Nanoparticle-tracking analysis (NTA) revealed that depleting mutated ß-catenin using either siRNA or shRNA increased the number of secreted particles (*Figure 2b*, *Figure 2—figure supplement 3*). The particle mean size (100–150 nm), unaltered by either approach, was consistent with the size of typical small EVs, such as exosomes (*Figure 2b*, *Figure 2—figure supplement 3*). To analyze the exosomal release in living HepG2 cells, we used a pH-sensitive reporter (CD63-pHluorin) that allowed the visualization of MVB–PM fusion by total internal reflection fluorescence (TIRF) microscopy (*Verweij et al., 2018*). The tetraspanin CD63 is a well-known exosome surface marker (*Jeppesen et al., 2019*; *Kowal et al., 2016*). In cells expressing CD63-pHluorin, we could detect multiple discrete increases in the fluorescence signal at the PM over the time suggesting ongoing fusion of CD63-pHluorin–positive acidic vesicles with the PM (*Figure 2—figure supplement 4*). We found that mutated ß-catenin depletion in HepG2 cells significantly increased the number of fusion events associated with secreted exosomes (*Figure 2c*, *Video 1* and *Video 2*). EVs released by HepG2 cells were next isolated by differential ultracentrifugation and characterized (*Figure 2d*). First, western-blot analysis of isolated EVs confirmed the expression of the CD63 exosomal marker. It is noteworthy that we observed an increase in CD63 expression after mutated ß-catenin depletion, suggesting an overall increase in the exosomal fraction (*Figure 2e*). This increased expression of CD63 was also confirmed with the proteomic analysis of HepG2 KD cells (*Figure 1f*). Moreover, the WT and the mutated forms of the ß-catenin protein were also found in HepG2-derived EVs, and the expression of mutated ß-catenin was decreased relative to the cells of origin (*Figure 2e*). Using transmission electron microscopy (TEM), we demonstrated that HepG2-derived EVs have the morphology (flat cup-shaped) and size (50–150 nm) of typical exosomes (*Yang et al., 2019*; *Figure 2f* and *Figure 2—figure supplement 5*). To better understand the regulation of exosome secretion by mutated ß-catenin, we next used TEM to visualize MVBs, the origin of exosomes. We observed a higher number of MVBs per cell and an increase in their diameter upon removal of mutated ß-catenin (*Figure 2g*). Taken together these results highlight that mutated ß-catenin represses exosome secretion.

## Mutated ß-catenin decreases Syndecan-4 and Rab27a expression in liver cancer cell lines

To explore the molecular aspect of the alteration of exosome secretion, we first attempted to confirm the data obtained in our transcriptomic and proteomic approaches. In HepG2 cells, we confirmed that transient (*Figure 3—figure supplement 1*) and stable (*Figure 3a–b*) mutated ß-catenin depletion increased gene and protein expression of both *SDC4* and *RAB27A*. Due to the lack of efficient antibodies against SDC4, we were unable to analyze SDC4 protein expression by Western-Blot. However, we were able to confirm the increased expression of Syndecan-4 and Rab27a by immunofluorescence (*Figure 3c–d*). We then extended these results to other liver cancer cell lines. We analyzed *SDC4*, *RAB27A*, *ARG1,* and *AXIN2* basal mRNA expression in five different liver cancer cell lines, each with a different *CTNNB1* mutational status, and identified positive correlations between *SDC4* and *RAB27A* with *ARG1* (ß-catenin-negative target) and negative correlations between *SDC4* and *RAB27A* with *AXIN2* (ß-catenin-positive target) (*Figure 3—figure supplement 2*). Upon analyzing the basal expression of Rab27a protein in these five cell lines, we identified a negative correlation between ß-catenin and Rab27a expression (*Figure 3—figure supplement 3*). In the two liver cancer

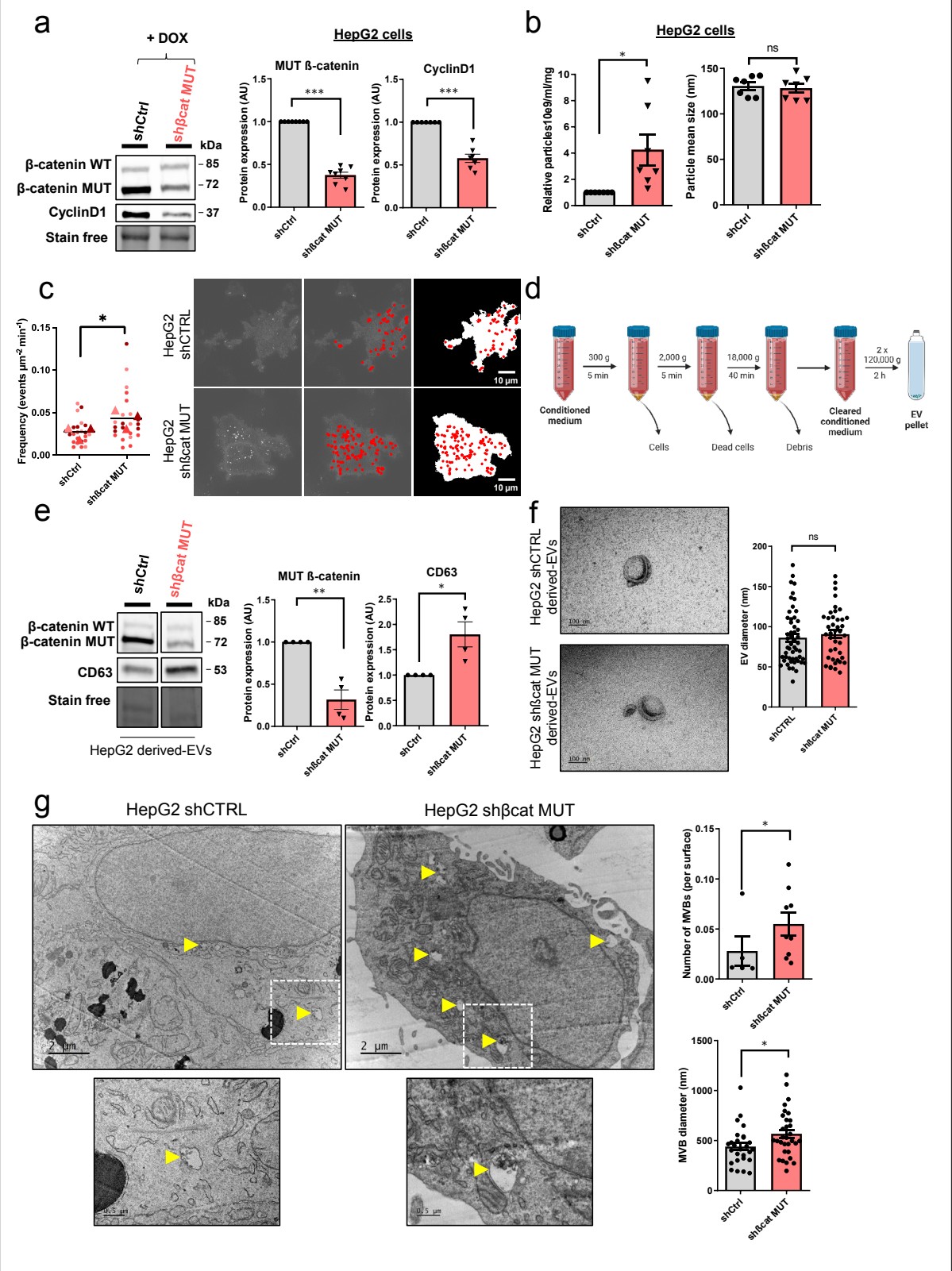

**Figure 2.** Mutated ß-catenin controls exosome secretion in HepG2 cells. (**a–g**) HepG2-shßCat MUT and HepG2-shCtrl cells were treated with doxycycline (DOX) to silence or not mutated ß-catenin. (**a**) Analysis of ß-catenin and CyclinD1 expression by western-blot. Stain-free was used as a loading control. Graphs show the quantification of seven independent experiments. (**b**) Nanoparticle tracking analysis of supernatant. Graphs show the quantification of seven independent experiments. (**c**) Quantification of CD63-pHluorin MVB–PM fusion events visualized by live total internal reflection

*Figure 2 continued on next page*

*Figure 2 continued*

fluorescence (TIRF) microscopy. Depicted data are representative of three independent experiments, each dot represents one cell. Images represent the cell mask (white) and red dots corresponding to fusion events. Scale bar: 10 µm. (**d**) Extracellular vesicles (EVs) isolation protocol. Created with BioRender.com, and published using a CC BY-NC-ND license with permission (**e**) Analysis of ß-catenin and CD63 expression in HepG2-derived EVs. Stain-free was used as a loading control. Graphs show the quantification of four independent experiments. (**f**) Transmission electron microscopy images of HepG2-derived EVs by close-up. Scale bar: 100 nm. The graph shows the diameter quantification of EVs (n=93). (**g**) Electron microscopy images of HepG2 shCtrl and shßcat MUT cells showing multivesicular bodies (MVBs) (yellow arrowheads). Scale bar: 2 µm (zoom: 500 nm). The graphs show the quantification of the number of multivesicular bodys (MVBs) per cell and the MVB diameter. (**a–g**) Results are expressed as Mean ± SEM, two-tailed Student's t-test analysis. *p<0.05; **p<0.01; ***p<0.001; ns, non-significant.

The online version of this article includes the following source data and figure supplement(s) for figure 2:

**Source data 1.** Original file for the Western blot analysis in *Figure 2a* (anti-ß-catenin).

**Source data 2.** Original file for the Western blot analysis in *Figure 2a* (anti-cyclin D1).

**Source data 3.** Original file for the Western blot analysis in *Figure 2a* (stain-free).

**Source data 4.** PDF containing *Figure 2a* and original scans of the relevant Western blot analysis (anti-ß-catenin, anti-cyclin D1, and stain-free) with highlighted bands and sample labels.

**Source data 5.** Original file for the Western blot analysis in *Figure 2e* (anti-ß-catenin).

**Source data 6.** Original file for the Western blot analysis in *Figure 2e* (anti-CD63).

**Source data 7.** Original file for the Western blot analysis in *Figure 2e* (stain-free).

**Source data 8.** PDF containing *Figure 2e* and original scans of the relevant Western blot analysis (anti-ß-catenin, anti-CD63, and stain- free) with highlighted bands and sample labels.

**Figure supplement 1.** Analysis of ß-catenin targets.

**Figure supplement 2.** Bile canaliculi in HepG2 cells treated with doxycycline to express either a control shRNA (shCtrl) or a shRNA targeting mutated ß-catenin (shßcat MUT).

**Figure supplement 3.** Nanoparticle tracking analysis of supernatant from HepG2 cell transfected with control (siCtrl) or mutated ß-catenin targeting (sißcat MUT) siRNA.

**Figure supplement 4.** Study of CD63-pHluorin MVB–PM fusion events.

**Figure supplement 5.** Transmission electron microscopy of extracellular vesicles (EVs).

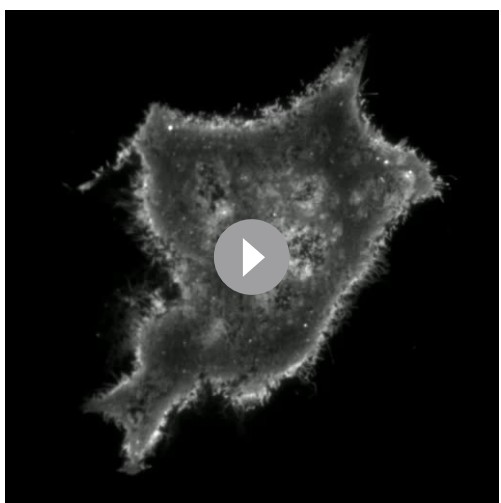

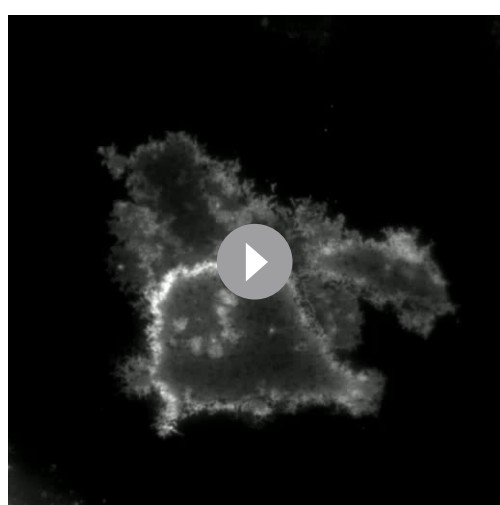

**Video 1.** HepG2-shCtrl. Cells were recorded under TIRF illumination on a microscope (IX83; Olympus) for 2 min. Videos show exocytic events (suddenly appearing fluorescent dots) corresponding to MVB-PM fusion.

https://elifesciences.org/articles/95191/figures#video1

**Video 2.** HepG2-shßCat MUT. Cells were recorded under TIRF illumination on a microscope (IX83; Olympus) for 2 min. Videos show exocytic events (suddenly appearing fluorescent dots) corresponding to MVB-PM fusion.

https://elifesciences.org/articles/95191/figures#video2

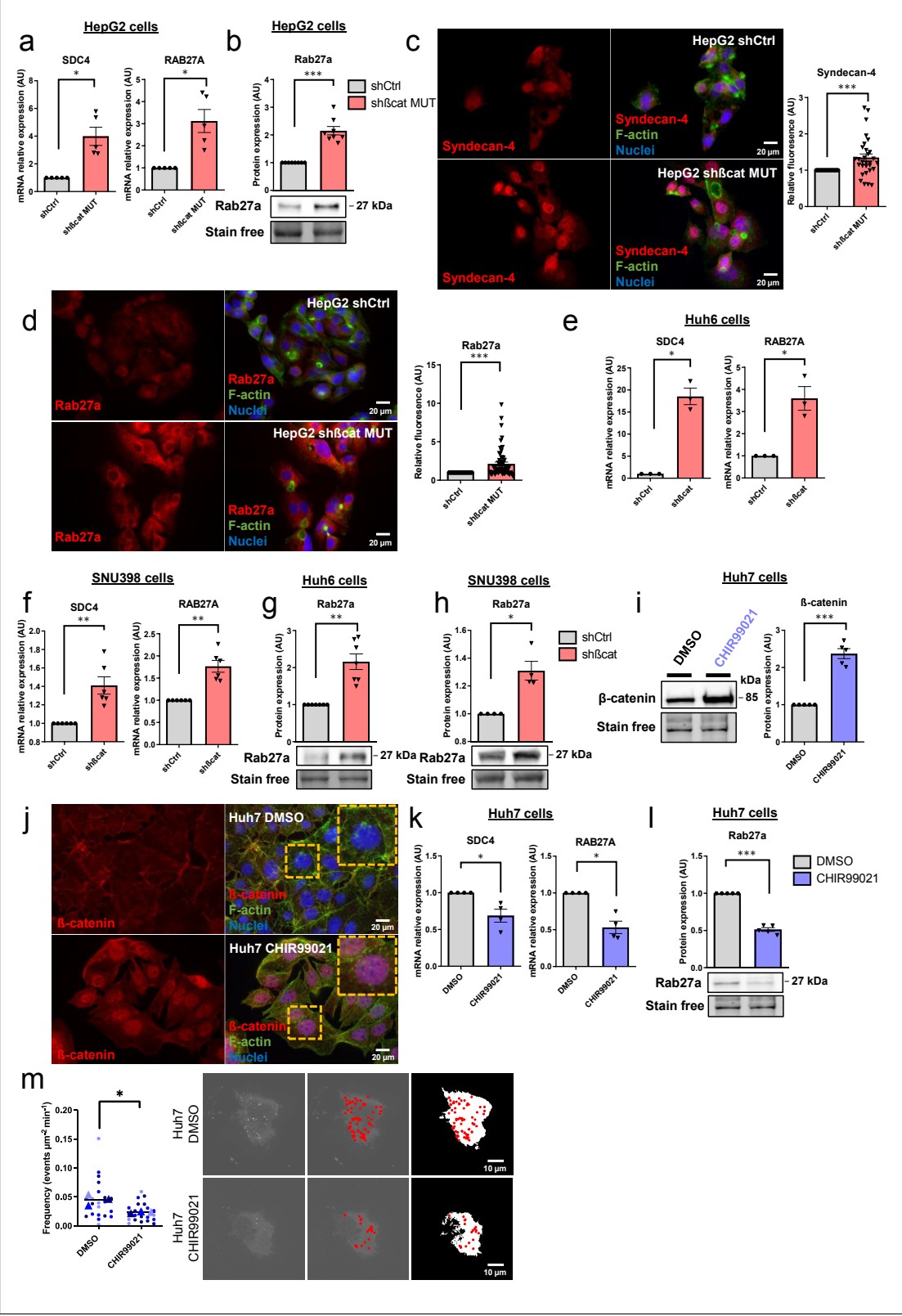

**Figure 3.** Activated ß-catenin represses *SDC4* and *RAB27A* expression in liver cancer cells. (**a–d**) HepG2 cells were treated with doxycycline (DOX) to express either a control shRNA (shCtrl) or a shRNA targeting mutated ß-catenin (shßcat MUT). (**a**) Analysis of *SDC4* and *RAB27A* mRNA expression by qRT-PCR. Graphs show the quantification. (**b**) Analysis of Rab27a protein expression by western-blot. Stain-free was used as a loading control. The graph shows the quantification. (**c–d**) Epifluorescence images of HepG2 shCtrl and shßcat MUT cells stained with Syndecan-4 and Rab27a antibodies

*Figure 3 continued on next page*

*Figure 3 continued*

(red), Phalloidin (green), Hoechst (blue). Scale bar: 20 µm. Graphs show the quantification of the fluorescence intensity per image and divided per nuclei. Depicted data are representative of three independent experiments, each dot represents one image. (**e**) Analysis of *SDC4* and *RAB27A* mRNA expression in Huh6 cells expressing either a control shRNA (shCtrl) or a shRNA targeting ß-catenin (shßcat) treated with DOX. The graph shows the quantification. (**f**) Analysis of *SDC4* and *RAB27A* mRNA expression in SNU398 cells expressing either a control shRNA (shCtrl) or a shRNA targeting ß-catenin (shßcat) treated with DOX. The graph shows the quantification. (**g–h**) Analysis of Rab27a protein expression in Huh6 (**g**) or SNU398 (**h**) cells expressing either a control shRNA (shCtrl) or a shRNA targeting ß-catenin (shßcat). Stain-free was used as a loading control. The graphs show the quantification. (**i–m**) Huh7 cells treated with DMSO or CHIR99021 (3 µM) for 48 hr. (**i**) Analysis of ß-catenin expression by western blot. The graph shows the quantification. (**j**) Epifluorescence images of cells stained with ß-catenin antibody (red), Phalloidin (green), Hoechst (blue). Scale bar: 20 µm. (**k**) Analysis of *SDC4* and *RAB27A* mRNA expression by qRT-PCR. The graphs show the quantification. (**l**) Analysis of Rab27a protein expression by western-blot. The graph shows the quantification. (**m**) Quantification of CD63-pHluorin MVB–PM fusion events visualized by live total internal reflection fluorescence (TIRF) microscopy. Depicted data are representative of three independent experiments; each dot represents one cell. Images represent the cell mask (white) and red dots corresponding to fusion events. (**a–m**) All graphs show the quantification of at least three independent experiments. Results are expressed as Mean ± SEM, two-tailed Student's t-test analysis. *$p<0.05$; **$p<0.01$; ***$p<0.001$.

The online version of this article includes the following source data and figure supplement(s) for figure 3:

**Source data 1.** Original file for the Western blot analysis in *Figure 3b* (anti-Rab27a).

**Source data 2.** Original file for the Western blot analysis in *Figure 3b* (stain-free).

**Source data 3.** PDF containing *Figure 3b* and original scans of the relevant Western blot analysis (anti-Rab27a and stain-free) with highlighted bands and sample labels.

**Source data 4.** Original file for the Western blot analysis in *Figure 3g* (anti-Rab27a).

**Source data 5.** Original file for the Western blot analysis in *Figure 3g* (stain-free).

**Source data 6.** PDF containing *Figure 3g* and original scans of the relevant Western blot analysis (anti-Rab27a and stain-free) with highlighted bands and sample labels.

**Source data 7.** Original file for the Western blot analysis in *Figure 3h* (anti-Rab27a).

**Source data 8.** Original file for the Western blot analysis in *Figure 3h* (stain- free).

**Source data 9.** PDF containing *Figure 3h* and original scans of the relevant Western blot analysis (anti-Rab27a and stain-free) with highlighted bands and sample labels.

**Source data 10.** Original file for the Western blot analysis in *Figure 3i* (anti-ß-catenin).

**Source data 11.** Original file for the Western blot analysis in *Figure 3i* (stain- free).

**Source data 12.** PDF containing *Figure 3i* and original scans of the relevant Western blot analysis (anti-ß-catenin and stain-free) with highlighted bands and sample labels.

**Source data 13.** Original file for the Western blot analysis in *Figure 3l* (anti-Rab27a).

**Source data 14.** Original file for the Western blot analysis in *Figure 3l* (stain-free).

**Source data 15.** PDF containing *Figure 3l* and original scans of the relevant Western blot analysis (anti-Rab27a and stain-free) with highlighted bands and sample labels.

**Figure supplement 1.** Study of SDC4 and Rab27a expression.

**Figure supplement 1—source data 1.** Original file for the Western blot analysis in *Figure 3—figure supplement 1* (anti-ß-catenin).

**Figure supplement 1—source data 2.** Original file for the Western blot analysis in *Figure 3—figure supplement 1* (anti-Rab27a).

**Figure supplement 1—source data 3.** Original file for the Western blot analysis in *Figure 3—figure supplement 1* (stain-free).

**Figure supplement 1—source data 4.** PDF containing *Figure 3—figure supplement 1* and original scans of the relevant Western blot analysis (anti-ß-catenin, anti-Rab27a, and stain-free) with highlighted bands and sample labels.

**Figure supplement 2.** Correlation of expression of *SDC4* and *RAB27A* with ß-catenin targets in liver cancer cell lines mutated (HepG2, SNU398, Huh6) or not (Huh7, Hep3B) for ß-catenin.

**Figure supplement 3.** Basal expression of ß-catenin and Rab27a in liver cancer cell lines mutated (HepG2, SNU398, Huh6) or not (Huh7, Hep3B) for ß-catenin.

**Figure supplement 3—source data 1.** Original file for the Western blot analysis in *Figure 3—figure supplement 3* (anti-ß-catenin).

**Figure supplement 3—source data 2.** Original file for the Western blot analysis in *Figure 3—figure supplement 3* (anti-Rab27a).

**Figure supplement 3—source data 3.** Original file for the Western blot analysis in *Figure 3—figure supplement 3* (stain-free).

**Figure supplement 3—source data 4.** PDF containing *Figure 3—figure supplement 3* and original scans of the relevant Western blot analysis (anti-ß-catenin, anti-Rab27a, and stain- free) with highlighted bands and sample labels.

**Figure supplement 4.** Huh6 and SNU398 model validation.

*Figure 3 continued on next page*

*Figure 3 continued*

**Figure supplement 4—source data 1.** Original file for the Western blot analysis in *Figure 3—figure supplement 4* (anti-ß-catenin) for Huh6 cells.

**Figure supplement 4—source data 2.** Original file for the Western blot analysis in *Figure 3—figure supplement 4* (stain-free) for Huh6 cells.

**Figure supplement 4—source data 3.** PDF containing *Figure 3—figure supplement 4* and original scans of the relevant Western blot analysis (anti-ß-catenin and stain- free) with highlighted bands and sample labels for Huh6 cells.

**Figure supplement 4—source data 4.** Original file for the Western blot analysis in *Figure 3—figure supplement 4* (anti-ß-catenin) for SNU398 cells.

**Figure supplement 4—source data 5.** Original file for the Western blot analysis in *Figure 3—figure supplement 4* (stain-free) for SNU398 cells.

**Figure supplement 4—source data 6.** PDF containing *Figure 3—figure supplement 4* and original scans of the relevant Western blot analysis (anti-ß-catenin and stain-free) with highlighted bands and sample labels for SNU398 cells.

**Figure supplement 5.** Huh6 and SNU398 model validation.

**Figure supplement 6.** Huh7 model validation.

cell lines bearing a ß-catenin point mutation (Huh6 and SNU398 cells), inducible shRNA strategy with doxycycline reduced ß-catenin protein expression by 63% and 49%, respectively (*Figure 3—figure supplement 4*). Associated with that reduction in ß-catenin protein expression we observed a decrease in *CCND1* and *AXIN2* mRNA expression and an increase in *ARG1* mRNA expression in both cell lines (*Figure 3—figure supplement 5*). As in HepG2 cells, ß-catenin depletion also increased *SDC4* and *RAB27A* gene expression and Rab27a protein expression in Huh6 and SNU398 cells treated with doxycycline (*Figure 3e–h*). In order to mimic the ß-catenin activation in a non-mutated HCC cell line, we treated Huh7 cells with a GSK3 inhibitor (CHIR99021) limiting ß-catenin phosphorylation and degradation. This CHIR99021 treatment increased ß-catenin protein expression (2.37-fold) (*Figure 3i*) and induced its translocation into the nucleus in Huh7 cells (*Figure 3j*) where concomitant deregulation of ß-catenin transcriptional targets was observed (*Figure 3—figure supplement 6*). Consistent with our earlier data using HepG2 cells, CHIR99021 treatment decreased *SDC4* and *RAB27A* gene expression, and Rab27a protein level in Huh7 cells (*Figure 3k–l*). Then, we used TIRF microscopy to analyze exosomal release in living Huh7 cells expressing CD63-pHluorin. We found that CHIR99021 treatment significantly decreased the number of fusion events in Huh7 cells (*Figure 3m*, *Video 3* and *Video 4*). All together, these results revealed that ß-catenin mutation/activation represses the expression of syndecan-4 and Rab27A, two proteins involved in exosome biogenesis, in association with repression of exosomal secretion.

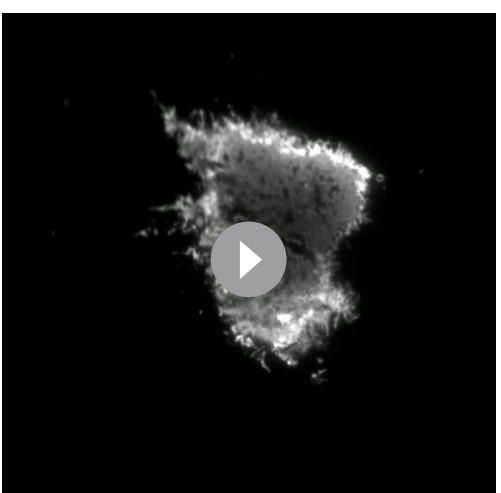

**Video 3.** Huh7 treated with DMSO. Cells were recorded under total internal reflection fluorescence (TIRF) illumination on a microscope (IX83; Olympus) for 2 min. Videos show exocytic events (suddenly appearing fluorescent dots) corresponding to MVB-PM fusion.

https://elifesciences.org/articles/95191/figures#video3

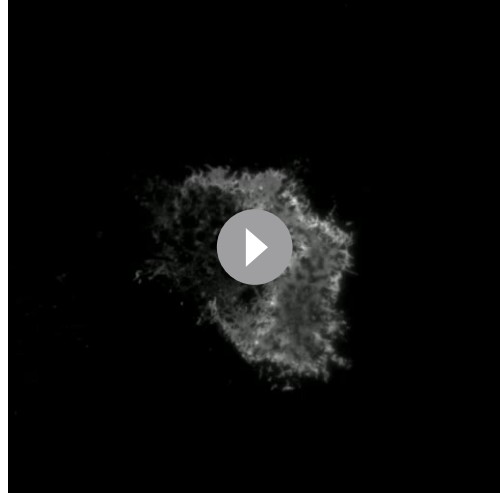

**Video 4.** CHIR99021 cells were transfected with CD63-pHluorin corresponding to *Figures 2c and 3m*. Cells were recorded under total internal reflection fluorescence (TIRF) illumination on a microscope (IX83; Olympus) for 2 min. Videos show exocytic events (suddenly appearing fluorescent dots) corresponding to MVB-PM fusion.

https://elifesciences.org/articles/95191/figures#video4

## Mutated ß-catenin promotes immune evasion in HCC cells through exosomes

Our next goal was to investigate the possible role of mutated ß-catenin in HCC immune escape. We used 3D spheroid models from two liver cancer cell lines (HepG2 and Huh7) and after 24 hr co-culture of spheroids with Peripheral Blood Mononuclear Cells (PBMC), immune cell infiltration was evaluated by flow cytometry (*Figure 4a–b*). First of all, we assessed the capability of PBMC to invade spheroids formed from Huh7 cells treated with CHIR9901. We observed a decrease in PBMC infiltration and an increase in tumor cell survival in Huh7 cells were treated with CHIR9901 compared to the results seen in the control condition (*Figure 4c*). Of note, we confirmed that CHIR99021 increased ß-catenin and decreased Rab27a protein expression in Huh7 spheroids with no impact on the cell viability (*Figure 4—figure supplement 1*). Next, we showed that the depletion of mutated ß-catenin protein enhanced PBMC infiltration in HepG2 cells associated with a decrease in tumor cell survival (*Figure 4d*). We verified that mutated ß-catenin protein was correctly depleted in HepG2 spheroids, with an increase in Rab27a protein expression and no impact on cell viability (*Figure 4—figure supplement 2*). Altogether, these results confirm the involvement of mutated ß-catenin in immune evasion. We next sought to investigate the role of exosomes in this process. In order to blunt exosome secretion, we used siRNA directed against Rab27a in spheroids formed with HepG2 cells depleted of mutated ß-catenin. Rab27a depletion decreased the number of secreted particles without any effect on cell viability in HepG2 shßcat MUT spheroids (*Figure 4—figure supplement 3*). Furthermore, depletion of Rab27a decreased PBMC infiltration associated with an increase of tumor cell survival (*Figure 4e*). Taken together, these results strongly suggest a determinant role for exosomes in immune cell infiltration.

## *CTNNB1* mutations are associated with low expression of exosomal biogenesis-associated genes in HCC patient samples

To determine whether markers of exosomal biogenesis are deregulated in HCC human samples, we assessed the expression of *SDC4* and *RAB27A* using public transcriptomic data from two different cohorts (*Figure 5a*: TGCA data sets from Cbioportal, *Figure 5b*; *Boyault et al., 2007*). In both sets of data, we showed that *SDC4* and *RAB27A* gene expressions were significantly reduced in *CTNNB1*-mutated HCCs compared to non-mutated tumors (*Figure 5a–b*). Furthermore, Pearson's correlation analysis of these two cohorts revealed that *RAB27A* and *SDC4* gene expression were positively correlated with the expression of *ARG1* and *PCK1* (ß-catenin-negative targets) (*Benhamouche et al., 2006*) and negatively correlated with the expression of *GLUL, CCND1, AXIN2, LGR5, FAT1,* and *BMP4* (ß-catenin-positive targets) (*Figure 5—figure supplement 1*). We then performed immunohistochemical analysis on 56 human HCC samples with or without *CTNNB1* mutations (*Figure 5—figure supplement 2*). We found that tumors strongly positive for glutamine synthetase (a well-known ß-catenin-positive target in the liver *Cadoret et al., 2002*), were correlated with low levels of Rab27a protein expression (*Figure 5c*), supporting the link between ß-catenin activation and regulation of exosomal secretion through a decrease in Rab27a expression.

## Discussion

ß-catenin is a main oncogene in tumors (*Kim and Jeong, 2019*) and several studies have reported a key role for mutated ß-catenin in immune escape in several types of cancer such as HCC, colorectal cancer, melanoma, and glioblastoma (*Sia et al., 2017*; *Spranger et al., 2015*; *Du et al., 2020*; *Wang et al., 2020*), but its mechanisms of action in this process remain to be elucidated. During the past decades, ß-catenin-mutated HCC have been classified at the molecular, histo-pathological, and clinical levels as liver cancers with specific features (*Hoshida et al., 2009*; *Shimada et al., 2019*; *Calderaro et al., 2017*). More recently, these tumors have been described as cold with low immune infiltrates and resistant to immunotherapy (*Akasu et al., 2021*; *Pinyol et al., 2019*; *Ruiz de Galarreta et al., 2019*; *Sia et al., 2017*).

In this study, we deciphered the role of mutated ß-catenin in cell communication between cancer cells and immune cells. Based on omic results, we hypothesized a role for EVs in the control of immune infiltrates in ß-catenin-mutated HCC. Following ISEV guidelines (*Théry et al., 2018*), we applied various techniques to address the impact of ß-catenin activation on EV secretion. Using NTA,

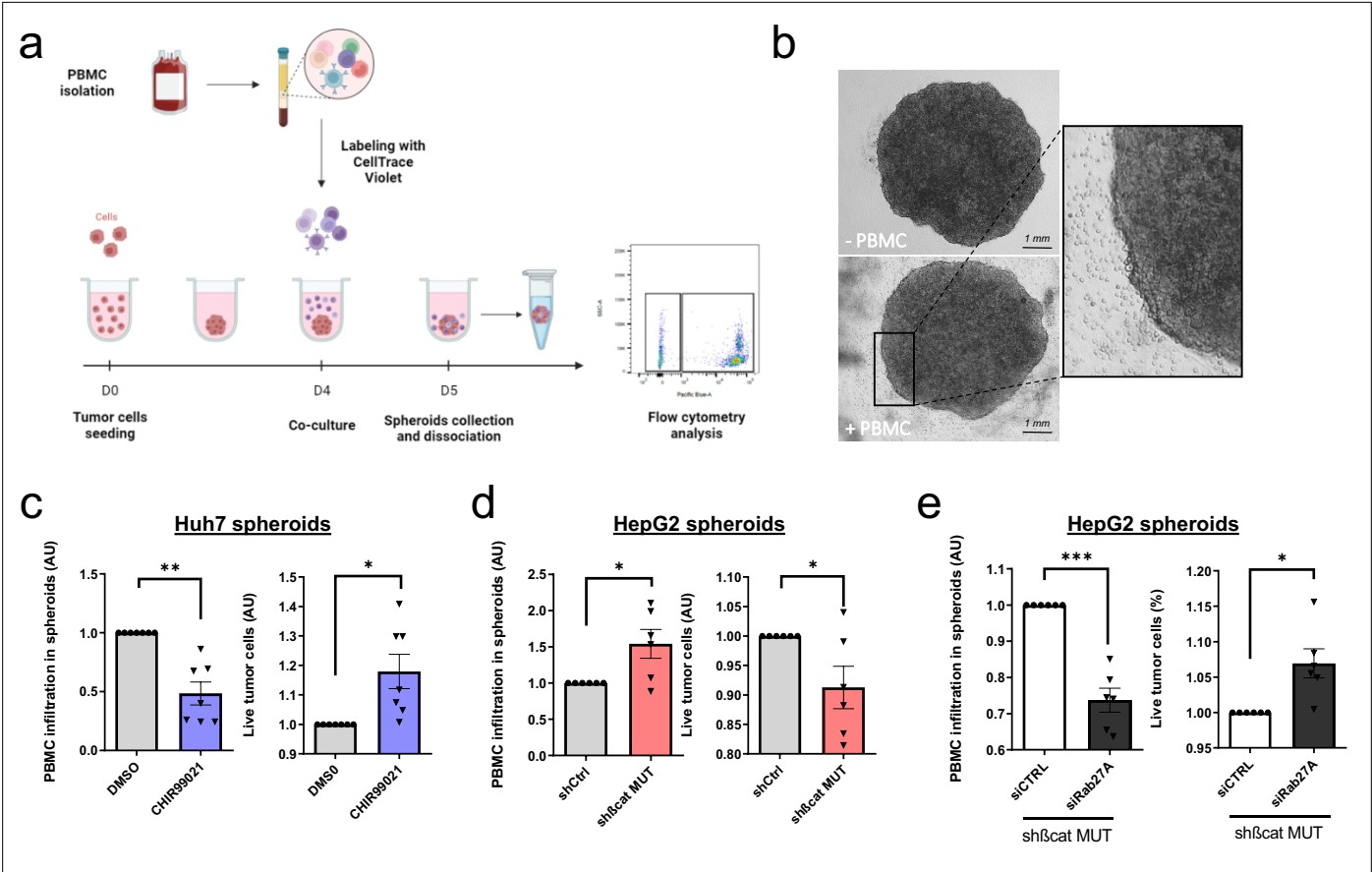

**Figure 4.** Activated ß-catenin represses immune infiltration in liver cancer spheroids through exosomes. (**a**) Peripheral blood mononuclear cells (PBMC) infiltration analysis protocol. Created with BioRender.com, and published using a CC BY-NC-ND license with permission. (**b**) Images of HepG2 spheroids expressing a control shRNA incubated or not with PBMC for 24 hr. (**c**) Analysis of PBMC infiltration and tumor cell survival in Huh7 spheroids treated with DMSO or CHIR99021. (**d**) Analysis of PBMC infiltration and tumor cell survival in HepG2 spheroids expressing control shRNA (shCtrl) or shRNA targeting mutated ß-catenin (shßcat MUT). (**e**) Analysis of PBMC infiltration and tumor cell survival in HepG2 spheroids co-expressing shRNA targeting mutated ß-catenin (shßcat MUT) and control siRNA (siCtrl) or siRNA targeting Rab27A (siRab27A). (**c–e**) Graphs show the quantification of six independent experiments. Results are expressed as Mean ± SEM, one or two-tailed Student's t-test analysis. *p<0.05; **p<0.01; ***p<0.001.

The online version of this article includes the following source data and figure supplement(s) for figure 4:

**Figure supplement 1.** Huh7 3D model validation.

**Figure supplement 1—source data 1.** Original file for the Western blot analysis in *Figure 4—figure supplement 1* (anti-ß-catenin).

**Figure supplement 1—source data 2.** Original file for the Western blot analysis in Figure 4-figure supplement 1 (anti Rab27a).

**Figure supplement 1—source data 3.** Original file for the Western blot analysis in Figure 4-figure supplement 1 (stain-free).

**Figure supplement 1—source data 4.** PDF containing *Figure 4—figure supplement 1* and original scans of the relevant Western blot analysis (anti-ß-catenin, anti-Rab27a, and stain-free) with highlighted bands and sample labels.

**Figure supplement 2.** HepG2 3D model validation.

**Figure supplement 2—source data 1.** Original file for the Western blot analysis in *Figure 4—figure supplement 2* (anti-ß-catenin).

**Figure supplement 2—source data 2.** Original file for the Western blot analysis in *Figure 4—figure supplement 2* (anti-Rab27a).

**Figure supplement 2—source data 3.** Original file for the Western blot analysis in *Figure 4—figure supplement 2* (stain-free).

**Figure supplement 2—source data 4.** PDF containing *Figure 4—figure supplement 2* and original scans of the relevant Western blot analysis (anti-ß-catenin, anti-Rab27a, and stain-free) with highlighted bands and sample labels.

**Figure supplement 3.** Silencing of Rab27A in HepG2 3D model.

**Figure supplement 3—source data 1.** Original file for the Western blot analysis in *Figure 4—figure supplement 3* (anti-Rab27a).

*Figure 4 continued on next page*

*Figure 4 continued*

**Figure supplement 3—source data 2.** Original file for the Western blot analysis in *Figure 4—figure supplement 3* (stain-free).

**Figure supplement 3—source data 3.** PDF containing *Figure 4—figure supplement 3* and original scans of the relevant Western blot analysis (anti-Rab27a and stain-free) with highlighted bands and sample labels.

we demonstrated that HepG2 cells silenced for mutated ß-catenin secrete more nanoparticles than control cells. Our different analyses and results (TIRF with the CD63 exosomal marker, TEM for the EVs shape and size, regulation of Rab27a expression) led us to take a strong interest in small EVs (i.e. exosomes), a subpopulation of EVs. Using real-time visualization of the release of CD63-enriched EVs we demonstrated that mutated ß-catenin silencing in HepG2 cells favors small EV secretion. The link

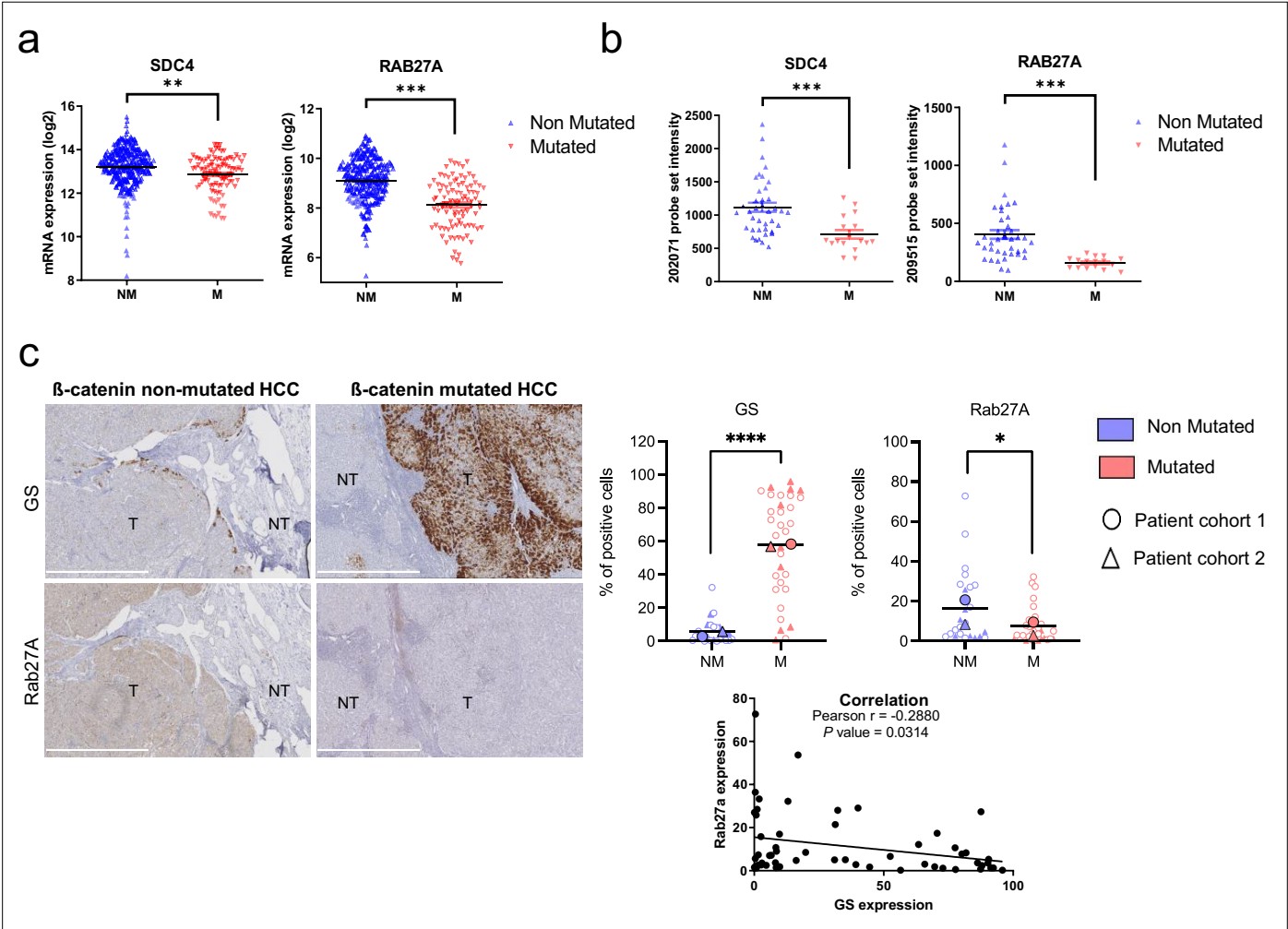

**Figure 5.** Downregulation of *SDC4* and *RAB27A* in human hepatocellular carcinoma (HCC) samples with *CTNNB1* mutations. (**a**) Analysis of *SDC4* and *RAB27A* mRNA expression in ß-catenin mutated (red) and non-mutated (blue) HCC (Cbioportal cohort, n=366). (**b**) Analysis of *SDC4* and *RAB27A* mRNA expression in ß-catenin mutated (red) and non-mutated (blue) HCC (Boyault *et al.* cohort, n=56). (**c**) Immunohistochemistry (IHC) analysis of glutamine synthetase (GS) and Rab27a in HCC samples presenting or not ß-catenin mutations. Scale bar: 1 mm. (T: tumoral, NT: non tumoral). The upper graphs show the quantification of the percentage of cells positive for GS or Rab27A. The analysis was split into two cohorts (circle and triangle). The lower graph shows the Pearson correlation between Rab27a and GS protein expression (n=56). Results are expressed as Mean ± SEM, two-tailed Student's t-test analysis, or Pearson correlation test. *p<0.05; **p<0.01; ***p<0.001; ****p<0.0001.

The online version of this article includes the following figure supplement(s) for figure 5:

**Figure supplement 1.** Correlation of expression of *SDC4* and *RAB27A* with various ß-catenin targets in human hepatocellular carcinoma (HCC) samples with *CTNNB1* mutations.

**Figure supplement 2.** Description of patient cohorts used for the immunohistochemical analysis.

between activation of ß-catenin and secretion of small EVs was further confirmed in ß-catenin non-mutated cells, Huh7 cells, treated with GSK3 inhibitor, suggesting that activation of the Wnt/ß-catenin pathway may result in a repression of the EV machinery in tumor cells. This link between ß-catenin and EVs was poorly explored in the literature and essentially restricted to the identification of ß-catenin as a common EV cargo (*Kalra et al., 2019*). For example, it was shown that tetraspanin expression reduces the cellular pool of β-catenin by enhancing the exosome-associated export of β-catenin from the cell and that this exosomal discharge of β-catenin downregulates the Wnt signaling pathway (*Chairoungdua et al., 2010*). More recently, using a genome-wide CRISPR/Cas9-mediated screening to identify genes involved in EV release, Lu et *al*. identified the Wnt signaling pathway in the top hits (*Lu et al., 2018*). Indeed, by this approach, a number of Wnt-signaling modulators appeared among the candidates as EV biogenesis regulators, which were further validated in chronic myelogenous leukemia K562 cells. Lu et *al*. also demonstrated that Wnt-mediated GSK3 inactivation, in K562 cells, regulated EV release in a Rab27a-dependent manner (*Lu et al., 2018*). Herein, we added new findings reporting that this regulation also occurs in solid tumors. Moreover, we demonstrated for the first time the specific involvement of the oncogenic ß-catenin in the regulation of exosome biogenesis.

At the molecular level, thanks to global approaches we found that oncogenic ß-catenin represses the expression of genes involved in the EV machinery. We indeed found that genes from the Rab (*RAB8B, RAB11A, RAB11B, RAB17, RAB27A*), Vamp (*VAMP2, VAMP5, VAMP8*) and Vps (*VPS4A, VPS4B, VPS16, VPS28*) families were upregulated upon removal of the oncogenic ß-catenin in HepG2 cells. Therefore, several trafficking steps could be impaired along the EV secretion pathway, the MVB biogenesis with the Vps, the transport and docking with the Rab, and the final step of membrane fusion with the Vamp (*van Niel et al., 2018*). Herein, we focused on *RAB27A* and *SDC4* genes, which were the most dysregulated in our transcriptomic analysis. We showed that mutated ß-catenin represses their gene expression in liver cancer cell lines. In non-mutated ß-catenin HCC cells, activation of ß-catenin showed that both gene and protein expression of Rab27a were decreased. We further found a correlation between the presence of *CTNNB1* mutations and the low expression of *RAB27A* and *SDC4* in human HCC patient samples. The way *RAB27A* and *SDC4* gene expression is repressed by ß-catenin remains to be explored. We described positive correlation between expression of *SDC4* and *RAB27A* and expression of negative transcriptional targets of ß-catenin, as well as a negative correlation with expression of positive targets in tumors. Even if the mechanism by which ß-catenin and LEF/TCF transcription factors led to target gene expression activation is well known (*Gest et al., 2023*; *Lustig et al., 2002*; *Nejak-Bowen and Monga, 2008*), it is not the case for the repression process. In melanoma, the transcription repressor ATF3 induced by the Wnt/ß-catenin pathway has been associated with *CCL4* repression (*Spranger et al., 2015*). Analysis of *SDC4* and *RAB27A* promoter sequences revealed the presence of consensus sequences of Lef1, Tcf4, and Atf3 binding (data not shown), suggesting that these newly identified target genes could be regulated at the transcriptional level by transcription factors known to associate with ß-catenin. In this line, *SDC4* was found to be a negative Tcf4 target gene in HCC mouse models (*Gougelet et al., 2014*).

Following the EV machinery alteration, we further addressed the impact of the ß-catenin-mediated repression of EV secretion on immune cell infiltration in tumors. Herein, the functional role of these tumor-derived exosomes in tumor immunity was studied using liver cancer 3D models and PBMC. Using siRNA depletion of Rab27a, we showed that mutated ß-catenin promotes a decrease of immune cell infiltration in micro-tumors through a decrease in exosome secretion. The small GTPase Rab27a has been implicated in several cancer progression mechanisms such as metastasis, cell invasion, or migration (*Li et al., 2018*). The oncogenic function of Rab27a is mainly due to its function as regulator of exosome secretion, which modulates cancer cell function and tumor microenvironment. In mammary carcinoma cells, the decrease of exosome secretion by Rab27a inhibition resulted in a diminution of primary tumor growth and lung metastatic dissemination (*Bobrie et al., 2012*). However, shRNA knockdown of Rab27a in a prostate cancer model revealed a dominant immunosuppressive role for EVs during antigen cross-presentation (*Salimu et al., 2017*). These data are inconsistent with our own, meaning so it seems that Rab27a-induced exosomal secretion could be a 'double-edged sword' in tumor progression. This could be due to the content of EVs. In fact, EVs are known to contain heterogeneous cargos that can play a critical role in immunomodulation (*Marar et al., 2021*). For example, the main immunosuppressive cytokine TGF-ß expressed by tumoral-derived EVs has been identified in Natural Killer cell function suppression (*Berchem et al., 2016*; *Viel et al., 2016*).

EVs released by tumoral cells can also carry several miRNAs mainly known to be involved in immune suppression (*Marar et al., 2021*). MiR-103a and Let-7a in tumor EVs promote M2-like polarization in lung and melanoma cancers (*Hsu et al., 2018*; *Park et al., 2019*). Interestingly, immunomodulatory proteins such as CSF-1, CCL2, FTH, FTL, and TGF-β chemokines have been identified in tumor exosomes (*Park et al., 2019*). The content of ß-catenin-regulated EVs remains to be explored to fully understand their function in the immunomodulation of the tumor microenvironment.

In conclusion, this study is the first description of the oncogenic ß-catenin as a key regulator of the exosome machinery. Moreover, it gave new insights regarding the regulation of tumor-derived exosome production by the ß-catenin pathway and their involvement in the immune escape observed in ß-catenin mutated HCC. Given that ß-catenin is also mutated in melanoma, endometrial, and colorectal cancers, these new findings may have a broader significance. Our data indicate that exosomes could be a promising biomarker and that restoring the tumor exosomal secretion of ß-catenin-mutated tumors may increase the communication between tumor and immune cells, and as a result, turn cold tumors hot.

# Materials and methods

## Key resources table

| Reagent type (species) or resource | Designation | Source or reference | Identifiers | Additional information |
|---|---|---|---|---|
| Cell line (*Homo sapiens*) | Hepatocellular carcinoma | ATCC | HepG2 | |
| Cell line (*Homo sapiens*) | Hepatocellular carcinoma | ATCC | Huh7 | |
| Cell line (*Homo sapiens*) | Hepatocellular carcinoma | ATCC | SNU398 | |
| Cell line (*Homo sapiens*) | Hepatoblastoma | Perret (Paris, France) | Huh6 | |
| Transfected construct (human) | AllStars negative-control siRNA | Qiagen | SI03650318 | |
| Transfected construct (human) | siRNA to ß-catenin | Eurofins Genomics *Gest et al., 2023* | | |
| Transfected construct (human) | siRNA to Rab27A | Sigma | MISSION EHU091501 | |
| Transfected construct (human) | pCMV-Sport6-CD36-pHluorin | Addgene | Plasmid# 130901 | TIRF experiment |
| Transfected construct (human) | Tet-pLKO-puro lentivirus vector | Addgene | plasmid# 21915 | Lentiviral construct to transfect and express the shRNA. |
| Peptide, recombinant protein | pLKO-Tet-On-shRNA-Control | Addgene | plasmid# 398398 | |
| Chemical compound, drug | GSK3 inhibitor CHIR99021 | Sigma | SML1046 | 3 µM |
| Biological sample (*Homo sapiens*) | PBMC | EFS | | Freshly isolated from human blood |
| Other | CellTrace Violet | Life Technologies | | 5 µM, FACS experiment to detect immune cells |
| Other | Propidium Iodide | Sigma | P4170 | 1/500 dilution, FACS experiment to detect dead cells |
| Sequence-based reagent | CCND1_F | *Gest et al., 2023* | PCR primers | CATCAAGTGTGACCCGGACTG |
| Sequence-based reagent | CCND1_R | *Gest et al., 2023* | PCR primers | CCTCCTCCTCAGTGGCCTTG |
| Sequence-based reagent | AXIN2_F | *Gest et al., 2023* | PCR primers | TGCTCTGTTTTGTCTTAAAGGTCTTGA |
| Sequence-based reagent | AXIN2_R | *Gest et al., 2023* | PCR primers | ACAGATCATCCCATCCAACACA |
| Sequence-based reagent | ARG1_F | *Gest et al., 2023* | PCR primers | GTGGACAGACTAGGAATTGGC |
| Sequence-based reagent | ARG1_R | *Gest et al., 2023* | PCR primers | TCCAGTCCGTCAACATCAAAAC |
| Sequence-based reagent | RAB27A_F | This paper | PCR primers | GAAGCCATAGCACTCGCAGAG |
| Sequence-based reagent | RAB27A_R | This paper | PCR primers | ATGACCATTTGATCGCACCA |
| Sequence-based reagent | SDC4_F | This paper | PCR primers | CGATGAGGATGTAGTGGGGC |

*Continued on next page*

*Continued*

| Reagent type (species) or resource | Designation | Source or reference | Identifiers | Additional information |
|---|---|---|---|---|
| Sequence-based reagent | SDC4_R | This paper | PCR primers | GACAACTTCAGGGCCGATCA |
| Sequence-based reagent | 18 S_F | *Gest et al., 2023* | PCR primers | GTAACCCGTTGAACCCCATT |
| Sequence-based reagent | 18 S _R | *Gest et al., 2023* | PCR primers | CCATCCAATCGGTAGTAGCG |
| Antibody | Anti-ß-catenin (mouse monoclonal) | BD Biosciences | 610154 | WB: 1/2000, IF: 1/400 |
| Antibody | Anti-Cyclin D1 (mouse monoclonal) | Santa Cruz | sc-20044 | WB: 1/1000 |
| Antibody | Anti-Rab27a (rabbit monoclonal) | Cell signaling | 69295 | WB: 1/1000, IF: 1/800, IHC: 1/100 pH6 |
| Antibody | Anti-CD63 (rabbit polyclonal) | Sigma | SAB4301607 | WB: 1/500 |
| Antibody | Anti-SDC4 (rabbit polyclonal) | Cell signaling | 12236 | IF: 1/200 |
| Other | Hoechst | Sigma | 34580 | IF: 1/1000, to detect nucleus |
| Other | 488 phalloidin | Interchim | YE5180 | IF: 1/200, to detect F-actin |
| Antibody | Anti-Glutamine synthetase (mouse monoclonal) | BD Biosciences | 610517 | IHC: 1/400 pH6 |

## Cell culture

Human HepG2 and Huh7 cell lines were grown in 4.5 g/L glucose Dulbecco's modified Eagle's Medium (DMEM, Gibco) supplemented with 10% FBS (Fetal Bovine Serum, Sigma). Human Huh6 cells were grown in 1 g/L glucose DMEM, (Gibco) supplemented with 10% FBS. Human SNU398 cells were grown in Roswell Park Memorial Institute medium (RPMI 1640, Gibco) supplemented with 10% FBS. Heat-inactivated FBS was used (30 min, 56 °C). All cell lines were cultured in a humidified atmosphere containing 5% $CO_2$ and 37 °C and mycoplasma contamination was checked regularly by PCR. All cell lines, except Huh6, were purchased from the American Type Culture Collection (ATCC). The Huh6 cell line was generously provided by C. Perret (Paris, France).

## Transfection

DNA transfections were performed using the Lipofectamine 3000 transfection Reagent according to the manufacturer's instructions (Invitrogen). The following plasmid was used: pCMV-Sport6-CD36-pHluorin (Plasmid#130901, Addgene). siRNA oligos were transfected into cells using Lipofectamine RNAiMax Reagent according to the manufacturer's protocol (Invitrogen). A reverse transfection was performed on day 1, a second forward transfection on day 2 and experiments were conducted on day 5. siRNA ß-catenin MUT (Eurofins Genomics) was previously reported (*Gest et al., 2023*), siRNA RAB27A (MISSION EHU091501, Sigma) targets human RAB27A. The AllStars negative-control siRNA from Qiagen was used as control siRNA.

## Lentiviral infection

HEK293T cells were seeded ($2.5 \times 10^6$) on a 10 cm plate coated with poly-L-lysine (Sigma) to obtain a confluence of 50–70% at the time of the infection. The next day, 600 µL of Opti-MEM (Gibco), 22 µL of Mirus LT1 transfection reagent (Mirus), and 4 µg of lentiviral plasmids of packaging vector (pPAX, pSD11) and lentiviral plasmid of interest were added on cells and incubated for 48 hr to allow the production of viruses. Then supernatant was harvested, centrifuged at 2500 rpm for 3 min to remove dead cells and debris, filtered, and used directly to infect cells or stored at –80 °C. After infection with lentiviral particles, cells were selected with puromycin treatment (2 µg/mL) for 1 week.

## shRNAs

The Tet-pLKO-puro lentivirus vector (plasmid#21915) and the control shRNA lentivirus vector (pLKO-Tet-On-shRNA-Control, plasmid#398398) were purchased from Addgene (Watertown, USA). The

construction of the two shRNA lentivirus vectors targeting the human ß-catenin was performed following the Tet-pLKO Manual given by Addgene (plasmid#21915). The shRNA sequences used were the same as previously described (*Gest et al., 2023*): the 5'-ACCAGTTGTGGTTAAGCTCTT-3 sequence to target the human ß-catenin in Huh-6 and SNU398 cells, and the 5'-TGTTAGTCACAA CTATCAAGA-3' sequence to target specifically the mutated form of ß-catenin in HepG2 cells. shRNAs were induced with doxycycline (1 µg/mL) in Huh6 cells for 5 days (2 treatments) and in HepG2 and SNU398 cells for 7 days (3 treatments).

## Drug treatment

Huh7 cells were treated with the GSK3 inhibitor CHIR99021 (3 µM, Sigma) for 48 hr before cell analysis. DMSO was used as control.

## Spheroid formation

Human Huh7 cells treated with either DMSO or CHIR99021 (3 µM; Sigma) for 48 hr were seeded (20,000 cells per well in 100 µL) in non-adherent conditions (ultra-low attachment 96 wells plate, Costar) in filtered 4.5 g/L glucose Dulbecco's modified Eagle's Medium (DMEM, Gibco) supplemented with 10% exosome free FBS. Human HepG2 cell line was transfected with either control shRNA or shRNA directed against mutated β-catenin (shBcat MUT) and induced with doxycycline (1 µg/mL, three rounds in one week). HepG2 shßcat MUT were also transfected with siRNA targeting Rab27a the day before the seeding for the reverse transfection (day 0) and the day of the seeding for the second forward transfection (day 1). 10,000 HepG2 cells were seeded per well in non-adherent conditions (ultra-low attachment 96 wells plate, Costar) in 100 µL filtrated 4.5 g/L glucose Dulbecco's modified Eagle's Medium (DMEM, Gibco) supplemented with 10% exosome free FBS for 96 hr. The induction of shRNA was maintained by adding doxycycline in each well (1 µg/mL in 50 µL) on the day of the seeding and 48 hr after.

## Peripheral blood mononuclear cell (PBMC) infiltration

After 96 hr of formation, Huh7 and HepG2 spheroids were co-cultured with CellTrace Violet (Life Technologies) labeled PBMC. Briefly, PBMC were isolated from donor blood (EFS Bordeaux) using a Ficoll (Eurobio Scientific) density gradient centrifugation. The obtained cells were then centrifuged (10 min, 1500 rpm) and washed several times in PBS 1 X. The remaining red blood cells were lysed by incubation in an ACK lysing buffer (Gibco), and the platelets were eliminated by centrifugation (10 min, 900 rpm). PBMC were then stained with Cell Trace Violet (5 µM, 20 min, 37 °C, $10^6$ cells/mL) and added to the wells (ratio tumor cells per spheroid:PBMC = 1:5, in 50 µL) for a 24 hr co-culture. After 24 hr, spheroids were collected, washed twice with PSB 1 X, and dissociated with trypsin 0.25%-EDTA (10 min, 37 °C, Gibco). Cells were then pelleted by centrifugation, resuspended in 200 µL of MACS buffer (PBS 1 X Dutscher, 0.5% FBS Eurobio Scientific, 0.4% EDTA 0.5 mM Euromedex), and labeled with Propidium Iodide (1/500 dilution, Sigma-Aldrich). The proportion of immune cells infiltrated in spheroids and tumor cells survival was then analyzed by flow cytometry (CantoII cytometer, BD Biosciences, Le Pont de Claix, France) and data analysis was performed with the FlowJo software (version 10.8.1).

**Table 1.** qPCR primers.

| Primer | Forward | Reverse |
|---|---|---|
| CCND1 | CATCAAGTGTGACCCGGACTG | CCTCCTCCTCAGTGGCCTTG |
| AXIN2 | TGCTCTGTTTTGTCTTAAAGGTCTTGA | ACAGATCATCCCATCCAACACA |
| ARG1 | GTGGACAGACTAGGAATTGGC | TCCAGTCCGTCAACATCAAAAC |
| RAB27A | GAAGCCATAGCACTCGCAGAG | ATGACCATTTGATCGCACCA |
| SDC4 | CGATGAGGATGTAGTGGGGC | GACAACTTCAGGGCCGATCA |
| 18 S | GTAACCCGTTGAACCCCATT | CCATCCAATCGGTAGTAGCG |

## qRT-PCR analysis

Total RNA was extracted from cells using the kit NucleoSpin RNA (Macherey-Nagel). RNA was then reverse transcribed using material from Thermo Fisher Scientific. qPCR was performed using the SYBR Green SuperMix (Quanta) using a C1000 Real-Time System (Bio-Rad). Data were normalized using the r18S gene as endogenous control and fold change was calculated using the comparative Ct method (-ddCt). All primers used are listed in *Table 1*.

## Western blot

Total proteins were extracted from cells or EVs using RIPA lysis buffer (0.1% SDS, 1% NP40, 0.15 M NaCl, 1% sodium deoxycholate, 25 mM Tris HCl pH 7.4) supplemented with protease and phosphatase inhibitors (Roche). Proteins were then denatured in Laemmli buffer (Bio-Rad) at 95 °C for 5 min. 40 µg of protein extract was loaded on 10% polyacrylamide gels (TGX Stain-Free FastCast, Bio-Rad). Membranes were blocked with 5% BSA-TBST (5% BSA Sigma, TBS 1 x Euromedex, 0,1% Tween20 Sigma) for 30 min and incubated with primary antibodies (diluted in 5% BSA-TBST) overnight at 4 °C. The following primary antibodies were used: ß-catenin (1:2000, mouse, 610154, BD Biosciences), Cyclin D1 (1:1000, mouse, sc-20044, Santa Cruz), Rab27a (1:1000, rabbit, 69295, Cell signaling), CD63 (1:500, rabbit, SAB430160, Sigma). Membranes were then incubated with secondary antibodies (1:5000 diluted in 5% BSA-TBST) for 30 min. The following secondary antibodies were used: IRDye 680CW conjugated goat anti-rabbit IgG (H&L) (LI-COR), IRDye 800CW conjugated goat anti-mouse IgG (H&L) (LI-COR). Acquisitions were performed using the ChemiDoc Imaging System (Bio-Rad). Intensities were measured using the ImageLab software (Bio-Rad).

## EVs isolation and characterization

Cells (2 million) were cultured in medium (7 mL) supplemented with EV-depleted FBS (obtained by ultracentrifugation for 16 hr at 120,000 g, using an Optima XPN-80 centrifuge with a 45.Ti rotor). EVs were isolated 72 hr after by differential centrifugation at 4 °C: 5 min at 300 g, 5 min at 2000 g, 40 min at 18,000 g and 120 min at 120,000 g. EV pellets were washed in PBS, centrifuged again at 120,000 g for 120 min, resuspended in PBS (or RIPA lysis buffer), and stored at –80 °C.

### Nanoparticle tracking analysis (NTA)

Number and size of particles were detected using the NS300 instrument (Malvern Panalytical Ltd., Malvern, UK) equipped with a 488 nm laser and a high-sensitivity scientific CMOS camera. Particles are tracked and sized based on Brownian motion and diffusion coefficient. Particles were diluted in PBS to obtain a concentration within the recommended range (30–120 particles/frame). Five 60 s videos were acquired for each sample with the following conditions: cell temperature 25 °C, syringe speed— 22 µL/s, camera level 14. Videos were subsequently analyzed with NanoSight Software NTA3.3.301 (Malvern Panalytical Ltd., Malvern, UK), which identified and tracked the center of each particle under Brownian motion to measure the average distance the particles moved on a frame-by-frame basis.

## Electron microscopy

### Cells

Cells were fixed in EM fixative solution (1.6% Ga in PB 0.1 M pH 7.4) for 1 hr at room temperature and then centrifuged. Cell pellets were embedded in 1% agarose. Samples were washed three times with PB 0.1 M and postfixed in 1.5% potassium ferrocyanide, 2% aqueous osmium tetroxide solution in PB 0.1 M for 1 hr at room temperature. Before and between each of the following incubations, samples were washed three times with distilled water. Samples were first incubated in fresh thiocarboxyhydrazide (TCH) solution for 20 min at room temperature. In a second postfixation step, a 2% osmium tetroxide solution was added to samples for 30 min at room temperature. Samples were then serially dehydrated in ethanol (35%, 35%, 50%, 70%, 80%, 90%, 100%, 100%, 100%; 2 min each), incubated in a solution of 50% ethanol and 50% Epon resin for 2 hr and incubated in 100% Epon resin overnight at room temperature. Epon resin was then renewed for another 4 hr of incubation. At the end, samples were embedded in a pure resin and cured by incubation at 60 °C for 48 hr. Ultramicrotomy was done with a Diatome 35° Diamond knife and a Leica EM-UC7 ultracut. Sections (70 nm thick) were collected on 200 mesh copper grid. Images were acquired using a transmission electron microscope (Hitachi H7650) at 80Kv equipped with an Orius Camera (Gatan) managed by Digital Micrograph software.

The quantification of the number of MVBs was performed by counting visible MVBs and then dividing per the cell surface using ImageJ software. The quantification of the MVB diameter was performed using ImageJ software.

### EVs

Cell supernatant fluid was concentrated using an Amicon Ultra-15 centrifugal filter from 6 mL to 500 µL at 4000 g during 30 min (100 k, Merk Millipore). EVs were isolated using the IZON qEV original 35 nm size exclusion column (Izon science) according to the manufacturer's instructions. EVs were then concentrated as mentioned previously and resuspended in an equal volume of 4% PFA in PBS (2% PFA final). 20 µL drops of the resuspended exosomes were then absorbed for 20 min at RT on hydrophilic carbon-coated grids. After PBS rinses, grids were placed on drops of 1% glutaraldehyde in PBS for 5 min. Several washes were done with distilled water, and grids were then transferred on a filtered drop of pH7 uranyl-oxalate solution, for 5 min in the dark. Grids were then transferred in drops of filtered 4% aqueous uranyl acetate/2,3 M methylcellulose (1 V for 9 V) on a petri dish covered with parafilm on ice for 10 min in the dark. Grids were then removed, one at time, with a stainless-steel loop, and excess fluid were blotted by gently touching the edge of the loop with a Whatman No. 1 filter paper. Grids were air-dried while still on the loop, torn off, and stored in a grid storage box. Observations were done with a TEM Hitachi H7650 at 80kV equipped with a Gatan Orius camera.

## Label-free quantitative proteomics

Three independent biological replicates were performed on total protein extracts from human HepG2 cells transfected with control siRNA or ß-catenin mutated siRNA. 10 µg of proteins were loaded on a 10% acrylamide SDS-PAGE gel and proteins were visualized by Colloidal Blue staining. Migration was stopped when samples had just entered the resolving gel and the unresolved region of the gel was cut into only one segment. The steps of sample preparation and protein digestion by the trypsin were performed as previously described (*Campion et al., 2021*). NanoLC-MS/MS analysis were performed using an Ultimate 3000 RSLC Nano-UPHLC system (Thermo Scientific, USA) coupled to a nanospray Orbitrap Fusion Lumos Tribrid Mass Spectrometer (Thermo Fisher Scientific, California, USA). Each peptide extract was loaded on a 300 µm ID × 5 mm PepMap C18 precolumn (Thermo Scientific, USA) at a flow rate of 10 µL/min. After a 3 min desalting step, peptides were separated on a 50 cm EasySpray column (75 µm ID, 2 µm C18 beads, 100 Å pore size, ES903, Thermo Fisher Scientific) with a 4–40% linear gradient of solvent B (0.1% formic acid in 80% ACN) in 57 min. The separation flow rate was set at 300 nL/min. The mass spectrometer operated in positive ion mode at a 2.0 kV needle voltage. Data were acquired using Xcalibur 4.4 software in a data-dependent mode. MS scans (m/z 375–1500) were recorded at a resolution of $R$=120,000 (@ m/z 200), a standard AGC target, and an injection time in automatic mode, followed by a top speed duty cycle of up to 3 s for MS/MS acquisition. Precursor ions (2–7 charge states) were isolated in the quadrupole with a mass window of 1.6 Th and fragmented with HCD@28% normalized collision energy. MS/MS data was acquired in the ion trap with rapid scan mode, a 20% normalized AGC target, and a maximum injection time in dynamic mode. Selected precursors were excluded for 60 s. Protein identification was done in Proteome Discoverer 2.5. Mascot 2.5 algorithm was used for protein identification in batch mode by searching against a Uniprot *Homo sapiens* database (75,793 entries, released September 3, 2020). Two missed enzyme cleavages were allowed for the trypsin. Mass tolerances in MS and MS/MS were set to 10 ppm and 0.6 Da. Oxidation (M) and acetylation (K) were searched as dynamic modifications and carbamidomethylation (C) as static modifications. Raw LC-MS/MS data were imported in Proline Studio for feature detection, alignment, and quantification (*Bouyssié et al., 2020*). Protein identification was accepted only with at least two specific peptides with a pretty rank = 1 and with a protein FDR value less than 1.0% calculated using the 'decoy' option in Mascot. Label-free quantification of MS1 level by extracted ion chromatograms (XIC) was carried out with parameters indicated previously. The normalization was carried out on median of ratios. The inference of missing values was applied with 5% of the background noise. The mass spectrometry proteomics data have been deposited to the ProteomeXchange Consortium via the PRIDE (*Deutsch et al., 2023*) partner repository with the dataset identifier: PXD043841.

## Immunofluorescence

Cultured cells were fixed with 4% PFA for 10 min, permeabilized with Triton-X100 0.1% (Sigma) for 10 min, and incubated in 4% BSA-PBS for 15 min to prevent non-specific binding. Cells were then incubated at room temperature with primary antibodies (diluted in 4% BSA-PBS) for 45 min and with secondary antibodies (1:200 diluted in 4% BSA-PBS) for 30 min. The following primary antibodies were used: ß-catenin (1:400, mouse, 610154, BD Biosciences), Rab27a (1:800, rabbit, 69295, Cell signaling), SDC4 (1:200, rabbit, 12236, Cell signaling). The following secondary antibodies were used: 547 H donkey anti-mouse IgG (H&L) (FP-SB4110-T, Interchim) and 547 H donkey anti-rabbit IgG (H&L) (FP-SB5110, Interchim). Nuclei were stained with Hoechst (1:1000, 34580, Sigma) and actin was stained with 488 phalloidin (1:200, FP-YE5180, Interchim). Coverslips were mounted on microscope slides using Fluoromount-G mounting media (SouthernBiotech) and were imaged under epifluorescence microscope (Zeiss) using the 63 X oil immersion objective and SP5 confocal microscope (Leica). Images were analyzed using ImageJ software.

## Immunohistochemistry

### Human samples

The total number of human HCC samples is 56 and has been provided by 2 cohorts obtained at different times (*Figure 5—figure supplement 2*). Experiments for each cohort have been performed by two different experimenters (cohort 1: NDS, cohort 2: IM) and using two different batches of Rab27a antibody.

### Experiment

Paraffin-embedded human HCC samples were cut into 3.5 µm thick tissue sections and processed for immunohistochemistry with the EnVision FLEX kit material (K800021-2, Agilent Dako) according to the manufacturer's instructions. Briefly, tissue sections were put on slides (TOM-1190, Matsunami) and placed on automatic staining racks at room temperature to be deparaffinized. Slides were immersed in a 75 °C pre-warmed Target Retrieval Solution Low pH (50 x) (K8005) in the Dako PTlink tank. The tank was then heated up to 95 °C, slides were incubated for 20 min and allowed to cool to 75 °C. Afterwards, tumor sections were soaked for 5–10 min in Dako Wash Buffer. Slides were placed on the automatic staining racks of the Dako Autostainer and peroxidase blocking reagent was added for 10 min. Sections were rinsed with Dako Wash Buffer and antibodies were applied for 45 min according to the following dilutions (Dako Antibody Diluent): Glutamine synthetase (1:400, pH6, mouse, 610517, BD Biosciences), Rab27a (1:100, pH6, rabbit, 69295, Cell Signalling), SDC4 (1:200, pH6, P11820-1-A, Proteintech). Two solutions were then applied to the slides with washing with Dako Wash Buffer before and between each application: HorseRadish Peroxidase for 20 min and substrate working solution. Finally, slides were rinsed with water and hematoxyline counterstaining was performed before mounting on coverslips (Eukitt classic mounting medium).

## Total internal reflection fluorescence (TIRF) imaging and analysis

Coverslips were placed in an imaging chamber, perfused at 37 °C with Hepes Buffer Saline (HBS) solution (135 mM NaCl, 5 mM KCl, 0.4 mM MgCl₂, 1.8 mM CaCl₂, 1 mM D-glucose, and 20 mM HEPES) and were adjusted to pH 7.4 and to 305 mOsm. Imaging was performed with an Olympus IX83 inverted microscope equipped for (TIRF) microscopy with a 100x, 1.49 NA objective (UAPON100X-OTIRF), a laser source (Cobolt Laser 06-DPL 473 nm, 100 mW), and an ILas2 illuminator (Gataca Systems) with a penetration depth set to 100 nm. Emitted fluorescence was filtered with a dichroic mirror (R405/488/561/635) and an emission filter (ET525/50 m, Chroma Technology) and recorded by an electron-multiplying charge-coupled device (EMCCD) camera (QuantEM 512 C, Teledyne Photometrics). Movies were acquired for 2 min at 10 Hz for exocytosis. To achieve good signal/noise ratio required for event detection and further analysis, fluorescence was bleached by high laser power illumination prior to acquisition of the full movie. MetaMorph 7.8 software was used for all acquisitions. Semi-automatic detection of exocytic events and their quantification were conducted using custom-made MATLAB scripts previously described (*Jullie et al., 2014*; *Sposini et al., 2017*; *Bakr et al., 2021*).

## Bioinformatic analysis

Public transcriptomic data of 366 HCC patient samples were provided from the Cancer Genome Atlas Research Network, downloaded from the cBioPortal site, and divided into two groups according to

the presence or not of *CTNNB1* hotspot mutations (respectively, 94 and 272 samples per group) by Genome data. Public transcriptomic data of 56 HCC patient samples were provided from the (*Boyault et al., 2007*) article and divided into two groups according to the presence or not of *CTNNB1* mutations (respectively, 17 and 39 samples per group).

## Statistical analysis

Data are expressed as mean ± SEM and are representative of at least three experiments. Statistical tests were carried out using GraphPad Prism software version 8.0.2 (GraphPad Software, San Diego, CA, USA). Statistical significance (p<0.05 or less) was determined using Student's *t-test* or analysis of variance (ANOVA). Values of p are indicated as follows: *p<0.05; **p<0.01; ***p<0.001; ****p<0.0001; ns, non-significant.

# Acknowledgements

We thank the FACSility, OneCell, Histopathologie, and Oncoprot facility platforms (TBMCore, US5, Bordeaux) for the help with flow cytometry, qPCR, immunohistochemistry, and mass spectrometry experiments. We acknowledge Cyril Dourthe for his help in the bioinformatic analysis. We thank Silvia Sposini (IINS, Bordeaux) for her help in the TIRF microscopy experiments. We thank Drs. Sara Basbous and Benjamin Bonnard (BRIC, Bordeaux) for helpful discussions on the project. We are grateful to Drs. Jean-Christophe Delpech, Liam Barry-Carroll (NutriNeuro Laboratory, Bordeaux), and Alexandre Favereaux (IINS, Bordeaux) for insightful discussions about EVs. We acknowledge the Bordeaux Imaging Center (Bordeaux, France) and the France BioImaging infrastructure supported by the French National Research Agency (ANR-10-INSB-04, 'Investments for the future'). We thank Professor William A Thomas (Colby-Sawyer College, New Hampshire) for proof-reading the manuscript. CD is supported by a PhD fellowship from both SIRIC BRIO and Région Nouvelle-Aquitaine. JV is supported by PhD scholarships from the French Ministry of Research (MENESR). This research was funded by La Fondation pour la Recherche Médicale to VM (équipe FRM 2018, grant number DEQ20180839586), La Ligue contre le Cancer (comité des Charentes et comité de la Gironde) to CB and l'AFEF (Association Française de l'Etude du Foie) to CB. The authors declare no competing financial interests.

# Additional information

### Funding

| Funder | Grant reference number | Author |
|---|---|---|
| Ligue Contre le Cancer | Comités Charentes et Gironde | Clotilde Billottet |
| Ministère de l'Education Nationale, de l'Enseignement Superieur et de la Recherche | PhD Student Fellowship | Justine Vaché |
| Association Française pour l'Etude du Foie | AAP research project 2022 | Clotilde Billottet |
| Fondation pour la Recherche Médicale | DEQ20180839586 | Violaine Moreau |

The funders had no role in study design, data collection and interpretation, or the decision to submit the work for publication.

### Author contributions

Camille Dantzer, Justine Vaché, Data curation, Formal analysis, Investigation, Methodology, Writing – original draft, Writing – review and editing; Aude Brunel, Data curation, Formal analysis; Isabelle Mahouche, Data curation, Formal analysis, Methodology; Anne-Aurélie Raymond, Jean-William Dupuy, Formal analysis, Methodology, Writing – original draft; Melina Petrel, Data curation, Methodology, Writing – original draft; Paulette Bioulac-Sage, Resources; David Perrais, Software, Formal analysis, Methodology; Nathalie Dugot-Senant, Methodology; Mireille Verdier, Barbara Bessette, Formal

analysis, Methodology; Clotilde Billottet, Conceptualization, Data curation, Formal analysis, Supervision, Funding acquisition, Investigation, Methodology, Writing – original draft, Writing – review and editing; Violaine Moreau, Conceptualization, Formal analysis, Supervision, Funding acquisition, Investigation, Methodology, Writing – original draft, Writing – review and editing

### Author ORCIDs
Jean-William Dupuy http://orcid.org/0000-0002-2448-4797
David Perrais http://orcid.org/0000-0002-5878-5408
Clotilde Billottet https://orcid.org/0000-0003-1819-984X
Violaine Moreau https://orcid.org/0000-0002-4513-7022

### Ethics
Human sample data collection, analysis and archiving are in accordance with French data protection laws. The data processing is also in accordance with the MR-004 Reference Methodology for which the Bordeaux university hospital and INSERM have signed a compliance agreement under number PV2022_353.

Reviewer #1 (Public Review): https://doi.org/10.7554/eLife.95191.3.sa1
Reviewer #2 (Public Review): https://doi.org/10.7554/eLife.95191.3.sa2
Reviewer #3 (Public Review): https://doi.org/10.7554/eLife.95191.3.sa3
Author response https://doi.org/10.7554/eLife.95191.3.sa4

---

## Additional files

### Supplementary files
• MDAR checklist

### Data availability
The mass spectrometry proteomics data have been deposited to the ProteomeXchange Consortium via the PRIDE partner repository with the dataset identifier: PXD043841.

The following dataset was generated:

| Author(s) | Year | Dataset title | Dataset URL | Database and Identifier |
|---|---|---|---|---|
| Dupuy JW, Billottet C | 2024 | Proteomic analysis of HepG2 cells after mutated ß-catenin depletion | https://www.ebi.ac.uk/pride/archive/projects/PXD043841 | PRIDE, PXD043841 |

The following previously published dataset was used:

| Author(s) | Year | Dataset title | Dataset URL | Database and Identifier |
|---|---|---|---|---|
| Gest C, Sena S, Dembele D, Moreau V | 2023 | Dual-function of beta-catenin in human tumor hepatocytes | https://www.ncbi.nlm.nih.gov/geo/query/acc.cgi?acc=GSE144107 | NCBI Gene Expression Omnibus, GSE144107 |

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
