## [Editor Report · eLife assessment]

Hepatocellular carcinoma (HCC) is a particularly aggressive form of cancer, with an increasing number of treatment options approved for use in patients over the past decade. However, the biology of HCC and identifiable therapeutic targets have not been as clear, even in the era of molecular oncology. Likewise, the cellular biology of HCC, including the role of intercellular communication, has not been well elucidated. In this **compelling** study, Dantzer et al. provide **fundamental** insight into the role of beta-catenin on intercellular communication occurring via extracellular vesicles, with implications for immune evasion in a cancer increasingly being treated using immuno-oncologic agents.

---

## [Referee Report · Reviewer #1 (Public Review)]

Summary:

This finding shows a connection between cancer associated beta-catenin mutations extracellular vesicle secretion. A link between the beta-catenin mutation and expression of trafficking and exocytosis machinery. They used a multidisciplinary approach to explore expression levels of relevant proteins and single particle imaging to directly explore the release of extracellular vesicles. These results suggest a role of extracellular vesicles in immune evasion in liver cancer with the role needing to be further explored in other forms of cancer. I find this work to be compelling and of strong significance.

Strengths:

This paper uses multidisciplinary methods to demonstrate a compelling role of beta-catenin mutations in suppressing EV secretion in tumors. The results and imaging are extremely convincing and compelling.

---

## [Referee Report · Reviewer #2 (Public Review)]

Summary:

Dantzer and colleagues are investigating the pivotal role of ß-catenin, a gene that undergoes mutation in various cancer cells, and its influence on promoting the evasion of immune cells. In their initial experiments, the authors developed a HepG2 mutated ß-catenin KD model, conducting transcriptional and proteomic analyses. The results revealed that the silencing of mutated ß-catenin in HepG2 cells led to an up-regulation in the expression of exosome biogenesis genes.

Furthermore, the researchers verified that these KD cells exhibited an increased production of exosomes, with the mutant form of ß-catenin concurrently decreasing the expression of SDC4 and Rab27a. Intriguingly, applying a GSK inhibitor to the cells resulted in reduced expression of SDC4 and Rab27a. Subsequent findings indicated that mutated ß-catenin actively facilitates immune escape through exosomes, and silencing exosome biogenesis correlates with a decrease in immune cell infiltration.

In a crucial clinical correlation, the study demonstrated that patients with ß-catenin mutations exhibited low levels of exosome biogenesis.

Strengths:

Overall, the data robustly supports the outlined conclusions, and the study is commendably designed and executed. However, there are a few suggestions for manuscript improvement.

Weaknesses: No weakness

---

## [Referee Report · Reviewer #3 (Public Review)]

Summary:

In this very important study by Dantzer et al., 'Emerging role of oncogenic b-catenin in exosome biogenesis as a driver of immune escape in hepatocellular carcinoma' the authors define a role for oncogenic b-catenin on exosome biology and explore the link between reduce exosome secretion and tumor immune cell evasion. Using transcriptional and proteomic analysis of hepatocellular carcinoma cells with either oncogenic or wildtype b-catenin the authors find that oncogenic b-catenin negatively regulates exosome biogenesis.

The authors can provide compelling evidence that oncogenic b-catenin in different hepatocellular carcinoma cells negatively regulates exosome biogenesis and secretion, by downregulation of, amongst others, SDC4 and RAB27A, two proteins involved in exosome biogenesis. The authors corroborate these results by inducing b-catenin activation using CHIR99021 in a hepatocarcinoma cell line with non-oncogenic bCatenin (Huh7 cells). The authors can further demonstrate convincingly that reduction in exosome release by hepatocarcinoma spheroids leads to a reduction in immune cell infiltration into the tumor spheroid.

Strengths:

This is a very important and well-conceived study, that appeals to a readership beyond the field of hepatocarcinoma. The authors demonstrate a compelling link between oncogenic bCatenin and exosome biogenesis. Their results are convincing and with well-designed control experiments. The authors included various complementary lines of investigation to verify their findings.

Weaknesses:

One limitation of this study is that the mechanistic relationship of exosome release and how they affect immune cells remains to be elucidated. In this context, the authors conclusions rest on the assumption that hepatocarcinoma immune evasion is based exclusively on the reduced number of exosomes. However, the authors do not analyze exosome composition between exosomes of wildtype and oncogenic background, which could be different.

---

## [Author Response]

The following is the authors’ response to the original reviews.

**Reviewer 1:**
While the role of Rab27 was strongly examined, the hits of the VAMP proteins were not explored in detail. I was wondering if the decrease in the presence of VAMPS directly suggests the final step of membrane fusion in the exocytosis of EVs is what is being impaired. Or if it is other trafficking steps along the EV secretion pathway.

We appreciate the relevance of this comment and we agree that the decrease of *VAMP gene expression* in the β-catenin-mutated HepG2 cells could suggest an impairment of the final membrane fusion step in exocytosis of EVs. We have therefore expanded this important point in the discussion (page 10). Indeed, we identified an upregulation of *VAMP2*, *VAMP5* and *VAMP8* expressions after mutated β-catenin depletion in the transcriptomic analysis of HepG2 cells. However, these proteins were not detected in the mass spectrometry analysis. Only VAMP3 and VAMP7 proteins were detected in the proteomic analysis without any variation. This is why we didn't focus on this trafficking step, but it could be interesting to explore it further in the future.

**Reviewer 2:**
(1) In Figure 1F, it is essential to investigate why mass spectrometry analysis indicated no significant changes in SDC4 levels.

We agree with the reviewer that indeed whereas we did observe a significant alteration of syndecan-4 expression at the mRNA level, we did not observe significant changes in syndecan-4 levels by mass spectrometry. One possible explanation is that heparan sulfate proteoglycans like syndecan-4 exhibit a high degree of structural heterogeneity due to the biosynthetic process that produces linear polysaccharides. This characteristic can alter the robustness of mass spectrometry analyses, leading to greater variability.

(2) Figure 2G lacks clarity in explaining how the quantification of MVBs (multivesicular bodies) was conducted.

We apologize for the lack in clarity in explaining how the quantification of MVBs was conducted in figure 2G. The *Materials and methods* section (part electron microscopy-cells, page 23) has been modified in order to emphasize this point.

(3) In Supplementary Figure 1F, there is a suggestion to highlight exosomes using arrowheads for enhanced clarity.

According to the reviewer’s suggestions, we added arrowheads on supplementary figure 1F in order to highlight the exosomes (page 16). This indeed improves clarity.

(4) Figure 3C prompts a question about the peculiar appearance of Actin staining in KD cells, requiring further investigation.

The peculiar appearance of this intense phalloidin staining between hepatocytes corresponds to bile canaliculi (BC), features of more differentiated HepG2 cells. As phalloidin-stained BC are very bright, this may diminish the visibility of other, thinner actin structures. We decided to change the image of KD cells for a more relevant one (new Figure 3C).

(5) An intriguing avenue for exploration is suggested in testing how the treatment of a GSK inhibitor on HepG2 cells might impact Rab27a and SDC4 expression.

We appreciate the relevance of the suggestion in testing how the treatment of a GSK inhibitor on HepG2 cells might impact Rab27a and SDC4 expression. According to the reviewer’s suggestions, experiments have been carried out and the data are presented in Author response image 1 below. In HepG2 cells, GSK inhibitor stabilized the wild-type β-catenin protein but surprisingly the mutated form of β-catenin is slightly decreased (Author response image 1A). Regarding the expression levels of both Rab27a and SDC4 mRNA, a small increase is observed (Author response image 1B). Rab27a protein is also increased upon the treatment with a GSK inhibitor on HepG2 cells (Author response image 1C). This increased in expression could be due to the decrease of the mutated form of β-catenin in HepG2 cells confirming that Rab27a and SDC4 are repressed by the mutated β-catenin.

**Author response image 1. sa4fig1:** Impact of a GSK inhibitor (CHIR99021) on Rab27a and syndecan-4 (SDC4) expressions in HepG2 cells. HepG2 cells were treated by 3 µM CHIR990221 or DMSO as control for 48h. (A) Western-blot (upper panel) and quantification (lower panel) of wild-type (WT) and mutated (MUT) β-catenin proteins in HepG2 cells treated with DMSO (control) or with CHIR990221. (B) qRT-PCR analysis of Rab27a and SDC4 expression in HepG2 cells treated with DMSO (control) or with CHIR990221. (C) Western-blot (left panel) and quantification (right panel) of Rab27a protein in HepG2 cells treated with DMSO (control) or with CHIR990221. *P<0.05

**Reviewer 3:**
(1) One limitation of this study is that the mechanistic relationship of exosome release and how they affect immune cells remains to be elucidated. In this context, the authors conclusions rest on the assumption that hepatocarcinoma immune evasion is based exclusively on the reduced number of exosomes. However, the authors do not analyze exosome composition between exosomes of wild type and oncogenic background, which could be different.

We agree that the mechanistic relationship of exosome release and how they affect immune cells remains to be elucidated. In the discussion we mentioned that the content of ß-catenin-regulated EVs remains to be explored to fully understand their function in the immunomodulation of the tumor microenvironment. In this line, we have ongoing experiments in order to analyse the exosomal content in term of proteins and microRNAs. According to our preliminary results, we are able to say that the exosome composition in knock-down mutated ß-catenin HepG2 cells compared to control HepG2 cells seems to be different suggesting not only an involvement of the number of exosomes in the immunomodulation but also of their content.

(2) The manuscript would benefit from minor language editing and the introduction from restructuring to enhance clarity.

The manuscript has now benefited from a language editing thanks to the Professor William A. Thomas (Colby-Sawyer College, New Hampshire). Acknowledgments have been modified (page 12) to thank the Professor William A. Thomas for proof- reading of the manuscript. The introduction has been also restructured and modified according to the reviewer's suggestions to enhance clarity (page 3).

(3) I believe that within the abstract, the authors mean 'defect' not 'default' in the sentence: Then, we demonstrated in 3D spheroid models that activation of β-catenin promotes a decrease of immune cell infiltration through a default in exosome secretion.

We apologize for the mistake between 'default' and 'defect' in the abstract. The abstract has been modified accordingly.

(4) Within the 'Introduction' part of the manuscript, the authors might consider reviewing and reorganizing the first paragraph for more clarity - I suggest leading with the first three sentences of the second paragraph (HCC is the most...) and then introducing b-catenin and the effects and implications of oncogenic ß-catenin in HCC.If the authors prefer the current structure of the 'Introduction', I would like to propose exchanging some of the wording:-In line 4: 'despite' instead of 'in front of'? Sentence: Thus, in front of the therapeutic revolution for cancers, with the emergence of immunotherapy and more particularly immune checkpoint inhibitors (anti-PD1, anti-PD-L1)-Additionally in line 7: In these tumors, the oncogenic β-catenin is able to set up a microenvironment that favors tumor progression notably by promoting immune escape. Here, 'establish' might be a better choice instead of 'set up' - In line 9 I suggest rephrasing the sentence: Few studies have reported that the defect of intercellular communication between cancer cells and immune cells is partly mediated by a decrease of chemokines production leading to a reduction of immune infiltrates.... and maybe adding a reference here.

The introduction has been altered accordingly. Thanks for these suggestions that helped us to improve our manuscript.